# Physiological insight into the conserved properties of *Caenorhabditis elegans* acid-sensing degenerin/epithelial sodium channels

Eva Kaulich[1] 🆔, Patrick T. N. McCubbin[1], William R. Schafer[1,2] 🆔 and Denise S. Walker[1] 🆔

[1]*Neurobiology Division, MRC Laboratory of Molecular Biology, Cambridge, UK*
[2]*Department of Biology, KU Leuven, Leuven, Belgium*

Handling Editors: Peying Fong & Morag Mansley

The peer review history is available in the Supporting Information section of this article (https://doi.org/10.1113/JP283238#support-information-section).

**Abstract** Acid-sensing ion channels (ASICs) are members of the diverse family of degenerin/epithelial sodium channels (DEG/ENaCs). They perform a wide range of physiological roles in healthy organisms, including in gut function and synaptic transmission, but also play important roles in disease, as acidosis is a hallmark of painful inflammatory and ischaemic

**Eva Kaulich** is a post-doctoral scientist in the Group of Professor William R. Schafer at the MRC Laboratory of Molecular Biology (Neurobiology Division). Her doctoral thesis, co-supervised by Dr Denise Walker, and current post-doctoral research focuses on the conserved and unique physiological properties of *C. elegans* acid-sensing ion channels, and their role in modulating neural circuits and driving behaviour.

This article was first published as a preprint. Kaulich E, McCubbin PTN, Schafer WR, Walker DS. 2022. Physiological insight into the conserved properties of Caenorhabditis elegans acid-sensing DEG/ENaCs. bioRxiv. https://doi.org/10.1101/2022.04.12.488049.

conditions. We performed a screen for acid sensitivity on all 30 subunits of the *Caenorhabditis elegans* DEG/ENaC family using two-electrode voltage clamp in *Xenopus* oocytes. We found two groups of acid-sensitive DEG/ENaCs characterised by being either inhibited or activated by increasing proton concentrations. Three of these acid-sensitive *C. elegans* DEG/ENaCs were activated by acidic pH, making them functionally similar to the vertebrate ASICs. We also identified three new members of the acid-inhibited DEG/ENaC group, giving a total of seven additional acid-sensitive channels. We observed sensitivity to the anti-hypertensive drug amiloride as well as modulation by the trace element zinc. Acid-sensitive DEG/ENaCs were found to be expressed in both neurons and non-neuronal tissue, highlighting the likely functional diversity of these channels. Our findings provide a framework to exploit the *C. elegans* channels as models to study the function of these acid-sensing channels *in vivo*, as well as to study them as potential targets for anti-helminthic drugs.

(Received 30 April 2022; accepted after revision 28 September 2022; first published online 6 October 2022)

**Corresponding authors** D. S. Walker: Neurobiology Division, MRC Laboratory of Molecular Biology, Francis Crick Avenue, Cambridge Biomedical Campus, Cambridge CB2 0QH, UK.    Email: dwalker@mrc-lmb.cam.ac.uk; E. Kaulich: Max Planck Institute for Brain Research, Max-von-Laue-Str. 4, 60438 Frankfurt am Main, Germany.    Email: eva.kaulich@brain.mpg.de; William Schafer: Neurobiology Division, MRC Laboratory of Molecular Biology, Francis Crick Avenue, Cambridge Biomedical Campus, Cambridge CB2 0QH, UK.    Email: wschafer@mrc-lmb.cam.ac.uk

**Abstract figure legend** Polar view of a phylogram of representative members of the DEG/ENaC superfamily, coloured according to phylum (Annelida, yellow; Arthropoda, light green; Chordata, dark green; Cnidaria, blue; Mollusca, dark purple; Nematoda, magenta; Placozoa, red). Construction of the phylogram is described in Fig. 1. Black and grey boxes indicate *C. elegans* acid-activated and acid-inhibited members, respectively.

**Key points**

- Acidosis plays many roles in healthy physiology, including synaptic transmission and gut function, but is also a key feature of inflammatory pain, ischaemia and many other conditions. Cells monitor acidosis of their surroundings via pH-sensing channels, including the acid-sensing ion channels (ASICs). These are members of the degenerin/epithelial sodium channel (DEG/ENaC) family, along with, as the name suggests, vertebrate ENaCs and degenerins of the roundworm *Caenorhabditis elegans*.
- By screening all 30 *C. elegans* DEG/ENaCs for pH dependence, we describe, for the first time, three acid-activated members, as well as three additional acid-inhibited channels.
- We surveyed both groups for sensitivity to amiloride and zinc; like their mammalian counterparts, their currents can be blocked, enhanced or unaffected by these modulators. Likewise, they exhibit diverse ion selectivity.
- Our findings underline the diversity of acid-sensitive DEG/ENaCs across species and provide a comparative resource for better understanding the molecular basis of their function.

## Introduction

Acidosis can occur under healthy physiological conditions, such as during synaptic transmission (Du et al., 2014), as well as being a hallmark of a wide range of pathologies. Cells monitor tissue acidosis through membrane proteins, including acid-sensing ion channels (ASICs) (Ortega-Ramirez et al., 2017; Vina et al., 2013). ASICs belong to the conserved family of degenerin/epithelial sodium channels (DEG/ENaC), non-voltage gated cation channels that are involved in a diverse range of cellular processes. As the name

indicates, the family also includes mammalian ENaCs and *Caenorhabditis elegans* degenerins, as well as *Drosophila* pickpockets (PPK) and an array of representatives from across animal phyla. Electrophysiological approaches, particularly using *Xenopus* oocytes, have played an essential role in establishing the physiology of channel properties of this diverse family (Canessa et al., 1993, 1994; Li et al., 2009; O'Brodovich et al., 1993; Schild et al., 1997; Zhang & Canessa, 2002).

Acid-sensing DEG/ENaC members across species can be classified into two groups, those activated and those inhibited by high proton concentrations. The former

group includes the mammalian ASICs (Waldmann et al., 1997; Zhang & Canessa, 2002), zebrafish zASICs (Chen et al., 2007), human ENaCs (Collier & Snyder, 2009) and *Drosophila* PPK1 (Boiko et al., 2012). Vertebrate ASICs are closed at neutral pH and generate proton-activated inward currents, which increase with decreasing extracellular pH. However, the precise properties depend on the subunit composition, with the half-activation pH varying from around 6.5 to 4.5 (Chen et al., 2007; Waldmann et al., 1997; Zhang & Canessa, 2002). Between them, they thus cover a significant pH range, with relevance to a diverse array of biological processes and contexts. ENaCs also exhibit pH dependence. For instance, the heteromeric human $\alpha\beta\gamma$ ENaC channel has its maximal current at pH 6 and minimal current at pH 8.5 (Collier & Snyder, 2009), correlating well with the pH range in the collecting duct of the kidney and other epithelia where ENaCs are expressed. Rat $\alpha\beta\gamma$ ENaC currents are not altered over the same pH range, whereas some amphibian ENaCs have much stronger pH sensitivity, highlighting variation between species (Collier & Snyder, 2009; Wichmann et al., 2019). Family members also vary in their kinetics; ASICs exhibit fast desensitisation, for example, with subunit-specific characteristics, whereas ENaC is constitutively active (see for reviews Gründer & Pusch, 2015; Hanukoglu, 2017).

The second group of acid-sensitive DEG/ENaC members display inward currents at neutral pH in the absence of additional stimulus, and are blocked by acidic pH. Until very recently, this group consisted of only three members, mouse ASIC5 (also called brain liver intestine Na$^+$ channel (BLINaC); Wiemuth & Grunder, 2010), *Trichoplax* TadNaC6 (Elkhatib et al., 2019) and *C. elegans* ACD-1 (Wang et al., 2008; Wang et al., 2012), to which we have added two more *C. elegans* channels, ACD-5 and the heteromeric FLR-1/ACD-3/DEL-5 channel (Kaulich et al., 2022).

The evidence linking ASICs to essential roles in neuronal health and disease makes them of particular interest, and tractable genetic model systems like *C. elegans* can facilitate insights into these processes. For instance, gain-of-function mutations in the *C. elegans* DEG-1, MEC-4 and UNC-8 subunits cause neuronal degeneration (Bianchi et al., 2004; Chalfie & Sulston, 1981; Wang et al., 2013). Introducing one of the causative mutations into human ASIC2 also causes neuronal cell death, suggesting that ASIC2 might also be involved in neurodegeneration (Waldmann et al., 1996). This hypothesis has received attention since ASIC2 is upregulated in patients with multiple sclerosis, an inflammatory neurodegenerative disease, and pharmacological blocking of ASICs can lessen clinical symptoms of inflammation and neuronal degeneration (Fazia et al., 2019; Friese et al., 2007). ASICs (and *C. elegans* DEG/ENaCs) are also targets of diverse non-steroidal anti-inflammatory drugs (NSAIDs) (Fechner et al., 2021; Voilley, 2004; Voilley et al., 2001),

highlighting their importance as targets for treating pain and inflammation (Dulai et al., 2021). Finally, ASICs could offer potential as targets for anti-helminthic drugs. A repertoire of compounds is currently used to treat or prevent parasitic nematode infections, including imidazothiazoles and macrocyclic lactones, which target nicotinic acetylcholine receptors and glutamate-gated chloride channels, respectively (Keiser & Utzinger, 2008; Williamson et al., 2009; Wolstenholme & Rogers, 2005). Widespread resistance is a critical threat to both agriculture and human health, so effective alternatives are urgently needed.

*C. elegans* DEG/ENaC expression has been described in various tissues including muscle, neurons, glia and intestinal epithelia, reflecting a wide variety of functions. For example, whereas *mec-4* and *mec-10* are expressed in mechanosensory neurons, *acd-1* and *delm-1* and *delm-2* are expressed in glia, *acd-5* and *flr-1* in the intestine, and *unc-105* in the body-wall muscle (Chalfie & Wolinsky, 1990; Driscoll & Chalfie, 1991; Han et al., 2013; Kaulich et al., 2022; Park & Horvitz, 1986a; Take-Uchi et al., 1998; Wang et al., 2008). However, despite the expansion and known diversity of *C. elegans* DEG/ENaCs, many members lack functional characterisation at the level of the channel, an obvious limitation when interpreting their *in vivo* function.

Aside from the acid-inactivated channels, ACD-1, ACD-5 and FLR-1/ACD-3/DEL-5 (Kaulich et al., 2022; Wang et al., 2008), the pH sensitivity of the remaining members was unknown. Therefore, we set out to perform a comprehensive screen for acid-sensitive *C. elegans* DEG/ENaC channel subunits. In addition to five subunits that form acid-inhibited homomers, we identified three acid-activated members. We demonstrated a diversity in the modulatory effect of amiloride, and also showed that the acid-activated members exhibit diversity in ion selectivity. Like their mammalian counterparts, their currents are blocked or potentiated by zinc, indicating further conservation of function across phyla. Thus, our study serves as the foundation for further screening for modulators of these channels and for understanding the molecular basis of diversity in DEG/ENaC function.

## Methods

### Protein sequences and alignment

Sequences of the DEG/ENaC superfamily were obtained from UniProt and combined into one file using SnapGene® (Dotmatics, San Diego, CA, USA, available at www.snapgene.com). Since removal of noisy or uncertain columns does not necessarily improve phylogenetic reconstruction (Tan et al., 2015), the complete amino acid sequences of the longest isoform (where applicable) were used for the phylogenetic estimation of protein

similarity. To address the issue that variable regions tend to be over-aligned, and consequently might lead to biases, the robust aligners PRANK (Loytynoja & Goldman, 2010; data not shown) and MAFFT (Katoh & Standley, 2013; Katoh et al., 2002) were used and confidence in the individual alignment columns was assessed using GUIDANCE2. Both alignments generated were similar. We used MAFFT as it allows re-adjustment alignment to reflect information from sequences aligned later (Larkin et al., 2007). The candidates were selected because their sequences were verified.

**Phylogram.** The DEG/ENaC superfamily protein sequences were aligned in MAFFT version 7 multiple alignment program using rough distance and average linkage UPGMA (unweighted pair group method with arithmetic mean) and the tree was visualised using iTOL (Ciccarelli et al., 2006; Kuraku et al., 2013; Letunic & Bork, 2019).

**Sequence similarity network.** The sequence similarity network (SSN) was generated using the web tool for SSNs for protein families (EFI-EST) developed by the Enzyme Function Initiative (EFI; efi.igb.illinois.edu/) (Gerlt et al., 2015; Gerlt, 2017; Zallot et al., 2018, 2019). Cytoscape was used to explore the SSN (Shannon et al., 2003). The EFI-EST webtools use NCBI BLAST and CD-HIT to generate SSNs. The computationally guided functional profiling tool uses the CGFP programs from the Balskus Lab (https://bitbucket.org/biobakery/cgfp/src) (Levin et al., 2017) and ShortBRED from the Huttenhower Lab (http://huttenhower.sph.harvard.edu/shortbred) (Kaminski et al., 2015). The data used in these analyses originated from the UniProt Consortium databases and the InterPro and ENA databases from EMBL-EBI.

***C. elegans* growth and maintenance.** Standard techniques were used for *C. elegans* strain maintenance (Brenner, 1974). All experiments were performed on hermaphrodite animals grown on *E. coli* OP50 at 22°C. Transgenic strains were generated by microinjection of plasmid DNA into the Bristol N2 wild-type strain (Mello et al., 1991).

### Molecular biology

The *Pdel-9::GFP* and *Pacd-2::GFP* transcriptional reporter plasmids, used for generating transgenic animals, were a kind gift from Professor Kyuhyung Kim's lab (Daegu Gyeongbuk Institute of Science & Technology (DGIST), Korea) and used the vector backbone pMC10 (M. Colosimo, unpublished). For *del-9*, 3113 bp upstream of the start codon was used; for *acd-2*, a 3004 bp fragment was used, encompassing 2526 bp upstream of the start codon of *acd-2* plus 478 bp downstream, i.e. including

part of the first intron. The *asic-1* promoter consisted of 3500 bp upstream of the start codon, fused to mKate2 in pDEST R4-R3 vector II. All reporter-fluorophore-fusion plasmids included the *unc-54* 3′UTR following the respective fluorophore gene. All plasmids, including the *Pasic-1* reporter fluorophore fusion and the KSM vector derivatives containing cDNAs for *Xenopus* oocyte expression, were assembled using NEBuilder® HiFi DNA Assembly Master Mix (cat. no. E2621L, New England Biolabs, Ipswich, MA, USA). *C. elegans* cDNA was obtained from growing N2 wild-type animals on 15 6-cm NGM plates until the food was diminished, and subsequently extracted and purified using the TRIzol Direct-zol RNA Miniprep (cat. no. R2051, Zymo Research, Irvine, CA, USA). cDNA was generated using the SuperScript$^{TM}$ III First-Strand Synthesis System (cat. no. 18080051, Thermo Fisher Scientific, Waltham, MA, USA). Primers were designed using SnapGene 5.0.4. (HiFi-Cloning of two fragments) based on the cDNA gene sequence found on wormbase.org (ordered from Integrated DNA Technologies Inc. (IDT), Leuven, Belgium) or Sigma-Aldrich (Merck Life Science UK Ltd, Gillingham, UK). The cDNA inserts were sub-cloned into the KSM vector under the control of the T7 promoter, with 5′ and 3′ untranslated regions (UTRs) of the *Xenopus* $\beta$-globin gene and a poly(A) tail. The forward primer AGATCTGGTTACCACTAAACCAGCC and reverse primer TGCAGGAATTCGATATCAAGCTTATCGATACC were used to amplify the KSM vector. NEB $T_m$ Calculator was used to determine annealing temperatures.

### Two-electrode voltage clamp in *Xenopus* oocytes

cRNA was synthesised using the mMessage mMachine T3 Transcription Kit (Thermo Fisher Scientific cat. no. AM1348), purified with GeneJET RNA Cleanup and Concentration Micro Kit (Thermo Fisher Scientific cat. no. K0841) and eluted in 15 µl RNase free water. *Xenopus laevis* oocytes of at least 1 mm in size were obtained from EcoCyte Bioscience (Dortmund, Germany). They were de-folliculated by collagenase treatment and maintained in standard 1× ND96 solution (96 mM NaCl, 2 mM MgCl$_2$, 5 mM HEPES, 2 mM KCl, 1.8 mM CaCl$_2$, pH 7.4). Oocytes were injected with 25 nl of cRNA solution at a total concentration of approximately 500 ng/µl (unless stated otherwise) using the Roboinject (MultiChannel Systems). Oocytes were kept at 16°C in 1× ND96 prior to two-electrode voltage clamp (TEVC) recordings. TEVC was performed 1–2 days post-injection at room temperature using the Roboocyte2 (MultiChannel Systems, Reutlingen, Germany). *Xenopus* oocytes were clamped at −60 mV, using ready-to-use Measuring Heads from MultiChannel Systems filled with 1.0 M KCl and 1.5 M potassium acetate (to reduce the build-up of salt crystals in the pipette). All channels were tested

using the Roboocyte2. For all current–voltage ($I$–$V$) step experiments, measurements were obtained in each solution once a steady-state current was achieved and the background leak current was subtracted.

As millimolar concentrations of $Ca^{2+}$ and other divalent ions except $Mg^{2+}$ can block ASIC currents (Paukert et al., 2004), $Ca^{2+}$-free buffers were used for substitution experiments of monovalent cations, adapted from a previous protocol (Hardege et al., 2015): 96 mM XCl, 1 mM $MgCl_2$, 5 mM HEPES, pH adjusted to 7.4 with XOH, where X was Na, K or Li. The osmolarity was checked and confirmed to be within the range of 210–240 mOsm or adjusted with D-glucose if necessary, as there is some effect on ENaC channel properties with lower osmolarity (Awayda & Subramanyam, 1998). Raw current $I$–$V$ curves for each individual oocyte were fitted to a linear regression line and the $x$-axis intercept was compared between solutions to calculate an average reversal potential ($E_{rev}$). Reversal potential shift ($\Delta E_{rev}$) when shifting from a NaCl to a KCl or LiCl solution was calculated for each individual oocyte. Estimation of internal ion concentrations has been a topic of debate for *Xenopus* oocytes, with estimates varying between 4 and 10 mM for $Na^+$ (Broer, 2010). Previous research has measured an internal $Na^+$ concentration of 14 mM for ENaC-expressing oocytes (Kusche-Vihrog et al., 2009), similar to other sodium channels at high $Na^+$ concentrations (Greeff & Kuhn, 2000). However, in contrast to ENaCs, which are sodium channels, the mammalian ASICs, the mollusc FaNaCs and *C. elegans* DEG/ENaCs show a high degree of variability in their ion permeability, especially for $K^+$ but also for divalent cations (Fechner et al., 2021; Grunder & Chen, 2010; Kashlan & Kleyman, 2011; Lingueglia et al., 1997; Vallee et al., 2021; Yang et al., 2017). As the Nernst equation only considers a single permeant ion, we used a modified Goldmann–Hodgkin–Katz equation to capture the permeability of our novel acid-sensing DEG/ENaCs to monovalent cations more accurately. Permeability ratios were calculated from the shift of the reversal potential of the $I$–$V$ relationship when $Na^+$ in the bath solution (at $pH_{50}$ concentrations for acid-activated channels and at pH 7.5 for acid-inhibited channels) was completely substituted by another ion, $X^+$ (which in this case was $K^+$ or $Li^+$); the permeability ratios for each of the channels were then calculated as previously described (Lynagh et al., 2020) where the ratios $P_{Na}/P_X$ were calculated using a modified Goldmann–Hodgkin–Katz equation:

$$P_X/P_{Na} = \exp[(V_{rev,Na} - V_{rev,X})F/RT],$$

where $V_{rev}$ is the reversal potential of a particular ion, $F$ is Faraday's constant, $R$ is the universal gas constant, and $T$ is the absolute temperature (in kelvins, 295.15 K); $F/RT = 0.03918$.

To test the responses to pH, the channel-expressing *Xenopus* oocytes were perfused with $1\times$ ND96 (using HEPES for buffering pH above 5.5 and MES for pH below); pH was adjusted with HCl ranging from pH 7.4 (standard pH of the 1X ND96 solution) to pH 4. For the $Zn^{2+}$ dose responses, a 1 M $ZnCl_2$ stock solution in water (cat. no. 229 997, Sigma-Aldrich) was diluted to the desired concentrations in $1\times$ ND96 buffer based on a previously established protocol (Chen et al., 2012). $Zn^{2+}$ was applied at increasing concentrations in the range 0.1 μM to 5 mM. For the amiloride dose responses and $I$–$V$ experiments, a 1 M amiloride stock solution in dimethyl sulfoxide (DMSO; cat. no. D12345, Thermo Fisher Scientific) was diluted to the desired concentrations in $1\times$ ND96 buffer either at pH 7.4 (for acid-inhibited channels) or $pH_{50}$ (for acid-activated channels). For the dose responses, amiloride was then applied at increasing concentrations, as indicated. Baseline subtraction and drift correction for all dose-responses was applied with Roboocyte2+ software (MultiChannel Systems). Analysis for pH, $Zn^{2+}$ and amiloride dose responses was performed as follows: currents were normalised to maximal currents ($I/I_{max}$) and best fitted with the Hill equation (variable slope) in GraphPad Prism version 9.0.2 (GraphPad Software Inc., San Diego, CA, USA). $N$ represents different oocytes from independent experiments (pooled together). All experiments were repeated on at least three different days (i.e. using different batches of oocytes).

### Confocal microscopy

Worms were mounted on 3% agar pads (in M9 buffer: 22 mM $KH_2PO_4$, 42 mM $Na_2HPO_4$, 85.5 mM NaCl, 1 mM $MgSO_4$) in a 3 μl drop of M9 buffer containing 25 mM sodium azide ($NaN_3$, Sigma-Aldrich). Images were acquired using a Leica Microsystems (Wetzlar, Germany) TCS SP8 STED 3X confocal microscope at $\times63$, $\times40$, or $\times20$ magnification and Z stacks were generated using Fiji (ImageJ) (Schneider et al., 2012).

### Availability of materials

All resources used in this study are shown in the key resource table (Table 1) below. *C. elegans* strains and plasmids generated for the purpose of this study are available upon request from wschafer@mrc-lmb.cam.ac.uk.

### Results

#### *C. elegans* DEG/ENaC protein sequences cluster with acid-sensing DEG/ENaCs from other species

*C. elegans* DEG/ENaCs are vastly expanded in number compared to their vertebrate members, with

**Table 1. Key resources (bacterial strains, chemicals, peptides, and recombinant proteins)**

| Reagent or resource | Source | Identifier | |
|---|---|---|---|
| **Bacterial strains** | | | |
| *E. coli* OP50 | Caenorhabditis Genetics Center | OP50 | |
| **Chemicals, peptides and recombinant proteins** | | | |
| Direct-zol RNA Miniprep | Zymo Research | | |
| SuperScript™ II Reverse Transcriptase | Thermo Fisher Scientific | | |
| pUCIDT-KAN-egas-1 cDNA | IDT | pEK308 | |
| pUCIDT-KAN-egas-2 cDNA | IDT | pEK309 | |
| pUCIDT-KAN-del-5_F9F3.4 | IDT | pEK155 | |
| **Experimental models: organism and strains** | | | |
| *C. elegans* var. Bristol N2 | Caenorhabditis Genetics Center | N2 (wild-type) | |
| ljEX1344 [Pacd-2::GFP (50ng/µl); Punc-122::GFP (50ng/µl)] | Current paper | AQ4647 | |
| ljEx1361 [Pdel-9::GFP (50ng/µl); Punc-122::GFP (50ng/µl)] | Current paper | AQ4672 | |
| ljEx1448 [Pasic-1::mKate2 (10ng/µl); Punc-122::GFP (50ng/µl)] | Current paper | AQ4840 | |
| **Primers for cloning promoters, gDNA and cDNA** | | | |
| KSM Hifiuni F | AGATCTGGTTACCACTAAACCAGCC | This paper | backbone for all KSM vectors if not stated otherwise |
| KSM hifiuni R | TGCAGGAATTCGATATCAAGCTTATCGATACC | | |
| acd-5 fragment_KSM_F | TTGGGCCCCTCGAGGTCGACATGGCGACGCGTAAGAAACC | This paper | pEK171 |
| acd-5 fragment_KSM_R | CTCCATTCGGGTGTTCTTGATTATGCTTCATGTATCACAGCTGGC | | |
| KSM vector_acd-5_F | AGGTTTCTTACGCGTCGCATGTCGACCTCGAGGGGCC | | |
| KSM vector_acd-5_F | CTGTGATACACATGAAGCATAATCAAGAAACACCCGAATGGAGTCTCT | | |
| acd-2_KSM_F | TTGGGCCCCTCGAGGTCGACATGCATCTCGAGGACGGTC | This paper | pEK214 |
| acd-2_KSM_R | CTCCATTCGGGTGTTCTTGATTAACGAGGAGACAAGGATGATGGTAAAG | | |
| KSM_acd-2_R | GGACCGTCCTCGAGATGCATGTCGACCTCGAGGGGC | | |
| KSM_acd-2_F | CATCCTTGTCTCCTCGTTAATCAAGAAACACCCGAATGGAGTCT | | |
| F28A12.1_ACD-4_KSM_F2 | TATCGAATTCCTGCAATGAAATAGAAAAGCAAAATTATCGTGCTTTGTATCTGTC | This paper | pEK252 |
| F28A12.1_ACD-4_KSM_R2 | GTGGTAACCAGATCTTCATAGTTTATAATAATTCCCAGATC | | |
| Y69H2.2_egas-3_KSM_F1 | CTTGATATCGAATTCCTGCAATGATTTTCCTGCTTTCCTCATATTCCC | This paper | pPM003 |
| Y69H2.2_egas-3_KSM_R1 | GTTTAGTGGTAACCAGATCTCTACTTTCATATTTCTGGCACAAAACCATAAACA | | |
| Y69H2.11_egas-1_KSM_F | GTTTAGTGGTAACCAGATCTTCACTTCCATCTTCCATACTTCTTACAACATAACGTGAATAG | This paper | pPM001 |
| Y69H2.11_egas-1_KSM_R | CTTGATATCGAATTCCTGCAATGCTACTATTCCTTCCTTCTTTTTCCCGG | | |

(Continued)

**Table 1. (Continued)**

| Reagent or resource | Source | Identifier |
|---|---|---|
| Y69H2.12_egas-2_KSM_F | CTTGATATCGAATTCCTGCAATGATTTTCCTGCTTTTCCTCATATTCCC | pPM002 |
| Y69H2.12_egas-2_KSM_R | GTGGTAACCAGATCTTCACTTTCTACAACATATTGTCAAAACTCCGAAC | |
| F55G1.13_egas-4_KSM_F | CTTGATATCGAATTCCTGCAATGTTGCTGCTATGGTTTTTCTTCCG | pPM004 |
| F55G1.13_egas-4_KSM_R | GTTTAGTGGTAACCAGATCTTCAAAGACGTTTGTTGAACAAAAGTATGAC | |
| T28B8.5_del-4_KSM_F | CTTGATATCGAATTCCTGCAATGGGTGTATTTTGGACCGGC | pEK230 |
| T28B8.5_del-4_KSM_R | GTTTAGTGGTAACCAGATCTTCAATCATTAGAAATGAGGCTTTGGTGGAAC | |
| F16F9.5_KSM_F | CGAATTCCTGCAGtacaaaattcaaaaaATGAATCGAAACCGC | pEK253 |
| F16F9.5_mec-10_KSM_R | GTTTAGTGGTAACCAGATCTTCAATACTCATTTGCAGCATTTCTC | |
| C47C12.6_deg-1_KSM_R | GTGGTAACCAGATCTTTATATTGATACGAAAGGCGTCGACTTTCGCC | pEK251 |
| C47C12.6_deg-1_KSM_F | CGAATTCCTGCAGCCCGGGGATCCACTAGTATGTGAACCATCACAGTAAAAC | |
| E02H4.1_del-1_KSM_F | GATATCGAATTCCTGCAATGGCAAGGAAGTATATTGATATTTAAAAAAATCAAAAATG | pEK329 |
| E02H4.1_del-1_KSM_R | GTTTAGTGGTAACCAGATCTTCAATTATTATTTGTGGATACTCCTTTTTCCGCA | |
| F59F3.4_del-5_KSM_F1 | CTTGATATCGAATTCCTGCAATGACGAGTGTCTCGTTTGGT | pEK228 |
| F59F3.4_del-5_KSM_R1 | GTTTAGTGGTAACCAGATCTTTAAAAATCATTCATAGGCATATTTTTGGTGAATGCT | |
| F_T28D9.7_del-10_KSM | CTTGATATCGAATTCCTGCAATGGTCCGCATGGCTGAG | pEK229 |
| R_T28D9.7_del-10_KSM | GTTTAGTGGTAACCAGATCTCTACACGTAAGAATGTTATCATCATCCTCTTCG | |
| R_del-9_C18B2.6_KSM | GTTTAGTGGTAACCAGATCTTCATATGGAGGCGTCGTTTCT | pEK219 |
| F_del-9_C18B2.6_KSM | CTTGATATCGAATTCCTGCAATGTACATGAATGAAATTTCCGAGAC | |
| R13A1.4c_KSM_F | TCGAATTCCTGCATGATCCAAAATATACATTTCCACGTCGC | pEK285 |
| R13A1.4c_unc-8c_KSM_R1 | GTGGTAACCAGATCTCTATTTGCTCATTAAACTCCTTGTTGATTCATTTG | |
| F25D1.4_degt-1_F | CTTGATATCGAATTCCTGCAATGCCTGAAAAAGAAGATCGAAGAC | pEK254 |
| F25D1.4_degt-1_R | GTTTAGTGGTAACCAGATCTTTATATAAAAAATTGTGGTTTAGGAATATATTACTTTTCTTTCGTTCAC | |
| F58G6.6a_del-2a_F | CTTGATATCGAATTCCTGCAATGTTCTGCTTTCTGCAGTTACCG | pEK247 |
| F58G6.6a_del-2a_R | GTTTAGTGGTAACCAGATCTTCAACATTGTCAGGCAAGTTTCTTCTGG | |
| F58G6.6b_del-2b_F | CTTGATATCGAATTCCTGCAATGAAAGGGCACACAGATTTTGATG | pEK248 |
| F58G6.6b_del-2b_R | GTTTAGTGGTAACCAGATCTTCAACATTGTCAGGCAAGTTTCTTCTGG | |
| asic-2_KSM_F | CTTGATATCGAATTCCTGCAATGCGCGGTGGCG | pEK264 |
| asic-2_KSM_R | GTTTAGTGGTAACCAGATCTTTATTTCTTCTTTTCTCCTCATCTCCTTATTCTCGA | |
| C27C12.5b_KSM_F1 | CCCTCGAGGTCGACGGATGACTGAAACTTCAAATTGCTCCAG | pEK207 |
| C27C12.5b_KSM_R1 | CTTAGAGACTCCATTCGGGTTTAGAAATCACAATTTCCGAGATACACAGAATTTCTTTT | |
| T28B8.5_del-4_KSM_F | CTTGATATCGAATTCCTGCAATGGGTGTATTTTGGACCGGC | pEK230 |
| T28B8.5_del-4_KSM_R | GTTTAGTGGTAACCAGATCTTCAATCATTAGAAATGAGGCTTTGGTGGAAC | |
| ZK770.1_KSM_F | TATCGAATTCCTATGGGAAAGAACAGCTTAAAACGGG | pEK234 |
| ZK770.1_KSM_R | GTAACCAGATCTATCAATTATCAAGATTAAACCCGTCTTTGTTTAAATTATAATCAG | |
| del-3_KSM_F | CTTGATATCGAATTCCTGCAATGTGGCTCCGAGGACTTTT | pEK236 |
| del-3_KSM_R | GTTTAGTGGTAACCAGATCTTTATGTGTCTCTGAAAGCTACATCTTGAC | |
| del-7_KSM_F | CTTGATATCGAATTCCTGCAATGAATTGTAGCTGTGGTCATCAAAACAG | pEK237 |
| del-7_KSM_R | GTTTAGTGGTAACCAGATCTTTATAGATCCATTTCGCGATTTTCTCGAA | |
| C24G7.4_KSM_F | TGATATCGAATTCCTGCAGCATGCATCTCGAGGACGGTC | pEK216 |
| C24G7.4_KSM_R1 | TCCATTCGGGTGTTCTTGAGTAACGAGGAGACAAGGATGGTGATGGTAAAGAGG | |

*(Continued)*

**Table 1. (Continued)**

| Reagent or resource | Source | Identifier |
|---|---|---|
| F02D10.5_KSM_F1 | AGCTTGATATCGAATTCCTGATGGAAAACGGAGACGGAAAGTG | This paper | pEK215 |
| F02D10.5_KSM_R1 | TTCTTGAGGCTGGTTAGTGTCAAATTAATTGTGATTTGAATATGGAGGATGTTGAAACT | This paper | |
| ACD-1_KSM_F | GCTTGATATCGAATTCCTGCATGGAGCCAACTTTATCTCCAAATTATCGAAATG | This paper | pEK216 |
| ACD-1_KSM_R | GGTTTAGTGGTAACCAGATCTTAATATTTTGAAAACTCGCGTTCCGGG | This paper | |
| UNC-105_KSM_F | GGTTTAGTGGTAACCAGATCTTTATGGTTCTTCTGGAGAGACTGGTCGATTACC | This paper | pEK385 |
| UNC-105_KSM_R | GCTTGATATCGAATTCCTGCAATGGAGAATGCGTCGTCAACTGCCC | This paper | |
| KSM_delm-1_F | CAAATTTACAAAAATATTAGTACCACTAAACCAGCCTCAAGAACACC | This paper | pEK226 |
| | | | |
| KSM_delm-1_F | TTTGATCTGACCACTAAACCAGCCTCAAGAACACC | | |
| C24G7.1_delm-2_KSM_F | GATATCGAATTCCTGCAGATGATTCCAACAATATGAAGCCAAAAAAACC | This paper | pEK384 |
| C24G7.1_delm-2_KSM_R | GTGGTAACCAGATCTCAGATCAAAACGCTCCTTGATCCGAC | | |
| mec-4_KSM_F | CTTGATATCGAATTCCTGCAATGTCATGGATGCAAAACCT | This paper | pEK266 |
| mec-4_KSM_R | GTTTAGTGGTAACCAGATCTTCAGAAAGATCCAGACGCAATTTCTTT | | |
| T21C9.3a_del-6a_F | GTTTAGTGGTAACCAGATCTTTATTTCTTATTAACAGTCTTTTATCAAAAGTATTTCCCTTCTTGAG | This paper | pEK249 |
| T21C9.3a_del-6a_R | CTTGATATCGAATTCCTGCAATGGGTGCCAAGGTAAAGGAT | This paper | |
| C11E4.3a_del-8_KSM_F1 | CTTGATATCGAATTCCTGCAATGCCGGATAAAAATCACAATTGCTG | This paper | pEK231 |
| C11E4.3a_del-8_KSM_F1 | GTTTAGTGGTAACCAGATCTTCACATGTTACTGTTGCCGAATAGTGG | | |
| **Recombinant DNA (Plasmids)** | | | |
| acd-5::KSM | Kaulich et al. (2022) | pEK171 |
| Pacd-2::GFP | Kyuhyung Kim Lab | pEK186 |
| Pdel-9::GFP | Kyuhyung Kim Lab | pEK199 |
| acd-3::KSM | Kaulich et al. (2022) | pEK207 |
| acd-2::KSM | Current paper | pEK214 |
| flr-1::KSM | Kaulich et al. (2022) | pEK215 |
| acd-1::KSM | Current paper | pEK216 |
| del-9::KSM | Current paper | pEK219 |
| del-5::KSM | Kaulich et al. (2022) | pEK228 |
| del-10::KSM | Current paper | pEK229 |
| del-4::KSM | Current paper | pEK230 |
| asic-1::KSM | Current paper | pEK234 |
| del-3::KSM | Current paper | pEK236 |
| del-7::KSM | Current paper | pEK237 |
| Pasic-1 (3.5kb)::mKate2 | Current paper | pEK240 |

*(Continued)*

**Table 1. (Continued)**

| Reagent or resource | Source | Identifier |
| --- | --- | --- |
| del-2a::KSM | Current paper | pEK247 |
| del-2b::KSM | Current paper | pEK248 |
| deg-1::KSM | Current paper | pEK251 |
| acd-4::KSM | Current paper | pEK252 |
| mec-10::KSM | Current paper | pEK253 |
| degt-1::KSM | Current paper | pEK254 |
| egas-1::KSM | Current paper | pPM001 |
| egas-2::KSM | Current paper | pPM002 |
| egas-3::KSM | Current paper | pPM003 |
| egas-4::KSM | Current paper | pPM004 |
| asic-2::KSM | Current paper | pEK264 |
| unc-8 (isoform c)::KSM | Current paper | pEK285 |
| del-1::KSM | Current paper | pEK396 |
| unc-105(isoform h)::KSM | Current paper | pEK385 |
| delm-1::KSM | Current paper | pEK226 |
| delm-2::KSM | Current paper | pEK384 |
| mec-4::KSM | Current paper | pEK266 |
| del-6 (isoform a)::KSM | Current paper | pEK249 |
| del-8 (isoform b)::KSM | Current paper | pEK231 |
| Software and algorithms | | |
| Prism | GraphPad Software Inc. | |
| Roboocyte2+ | Multichannel Systems Inc. | |
| SnapGene | Dotmatics | |
| GUIDANCE2 | Sela et al. (2015) | |
| MAFFT | Katoh et al. (2002), Katoh & Standley (2013) | |
| iTOL | Ciccarelli et al. (2006), Kuraku et al. (2013), Letunic & Bork (2019) | |
| EFI-EST | Enzyme Function Initiative (efi.igb.illinois.edu/) | |
| Other | | |
| Roboocyte2 | Multichannel Systems Inc. | |
| Roboinject | Multichannel Systems Inc. | |

30 subunit-encoding genes (compared to 8–9 in vertebrates). As the phylogram and the sequence similarity network in Figs 1 and 2 show, they form distinct homology groups, with the vertebrate ENaCs and ASICs clustering separately (see also Table 2 for sequences used). However, it is not clear whether these clusters share functional characteristics. Indeed, the *C. elegans* DEG/ENaCs are the only group with members in multiple clusters. The vertebrate ASICs closely cluster with the TadNaCs from *Trichoplax adhaerens*, a primitive multicellular animal lacking any internal organs and neurons, and both groups have members that are sensitive to changes in proton concentration (Elkhatib et al., 2019; Zhang & Canessa, 2002).

### The *C. elegans* DEG/ENaCs include both acid-inhibited and acid-activated channels

To identify other acid-sensitive *C. elegans* DEG/ENaC members, we performed a preliminary screen for pH-sensitive subunits using two-electrode voltage clamp (TEVC) in *Xenopus* oocytes injected with cRNA,

systematically testing all 30 *C. elegans* DEG/ENaCs (Fig. 3; for FLR-1, ACD-3 and DEL-5 see Kaulich et al. (2022)). We used *Xenopus* oocytes because they express relatively few endogenous channels and no ASICs that could interfere with our recordings (Weber, 1999). Previous research has also shown that oocytes exposed to an acid solution, which had been acidified with HCl, do not show a significant or reproducible increase in inward current, and currents of water-injected oocytes are not reduced by alkaline pH or enhanced by acidic conditions (see, for example, Collier & Snyder, 2009; Wichmann et al., 2019; Zhang & Canessa, 2002) making them ideal for screening for acid sensitivity. For this initial screen, oocytes were perfused with pH 7.4 and pH 4 solutions (note that at this stage we did not verify currents by addition of a channel blocker, since several family members are known to be insensitive to, or potentiated by, amiloride or its derivatives and most of the *C. elegans* members remained uncharacterised; Adams et al., 1999; Fechner et al., 2021; Li, Yu et al., 2011; Matasic et al., 2021). We identified two groups of candidate acid-sensitive *C. elegans* DEG/ENaCs (Fig. 3*C* and *D*) that we went on to characterise in more detail. One group included

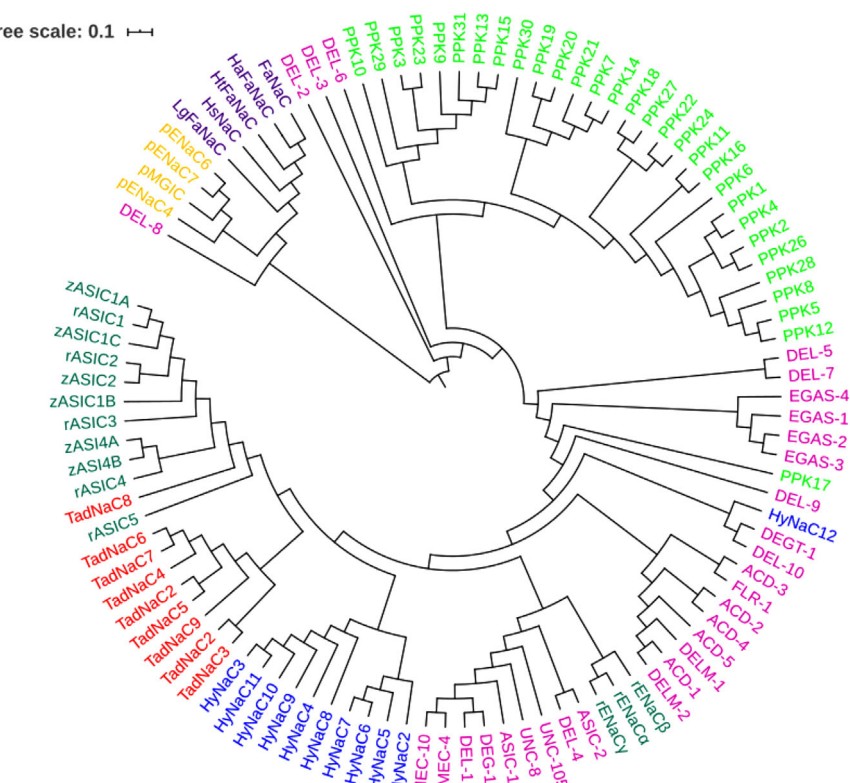

Annelida (*Platynereis dumerilii*)
Arthropoda (*Drosophila melanogaster*)
Chordata (*Danio rerio; Rattus norvegicus*)
Cnidaria (*Hydra vulgaris*)
Mollusca (*Helix aspersa, Helisoma trivolvis, Aplysia kurodai, Lymnaea stagnalis, Helix stagnalis*)
Nematoda (*Caenorhabditis elegans*)
Placozoa (*Trichoplax adhaerens*)

Tree scale: 0.1

**Figure 1. Phylogram of the DEG/ENaC family**
Polar view of a phylogram of the DEG/ENaC superfamily; protein sequences were aligned in MAFFT version 7 (see Methods for details). Tree scale represents the amount of genetic change. The data used to build these networks originate from the UniProt Consortium databases and the InterPro and ENA databases from EMBL-EBI. Colouring is according to phyla, as indicated. Accession numbers can be found in Table 2. Lower case letters indicate species of one group, i.e. for Chordata, z indicates zebrafish (*Danio rerio*) and r indicates rat (*Rattus norvegicus*), for Annelida p indicates *Platynereis dumerilii*, Lg for *Lottia gigantean*, Ls for *Lymnaea stagnalis*, Ha for *Helix aspersa* and Ht for *Helisoma trivolvis*. *C. elegans* names reflect homology or phenotype of mutants. ACD, acid-sensitive degenerin; ASIC, acid-sensing ion channel; DEG, degenerin; DEL, degenerin like; DELM, degenerin linked to mechanosensation; EGAS, EGF plus ASC domain ion channel; ENaC, epithelial sodium channel; FaNaC, FMRFamide-gated sodium channel; FLR, fluoride resistant; ; HyNaC, *Hydra* sodium channel; MEC, mechanosensory abnormality; PPK, Pickpocket; TadNaC, *Trichoplax adhaerens* sodium channel; UNC, uncoordinated. [Colour figure can be viewed at wileyonlinelibrary.com]

the previously characterised ACD-1 (Wang et al., 2008) and ACD-5 (Kaulich et al., 2022), showing currents at neutral pH (pH 7.4) that were decreased at low pH (pH 4). We identified three additional potential members, DEL-4, DELM-1 and UNC-105 (isoform h), fitting this profile (Fig. 3*D*; 'Statistical summary document'), and which we will refer to here as acid-inhibited channels. In this initial screen, DELM-1-expressing oocytes were not statistically significantly different from water-injected controls; this reflected a variability between oocytes. A portion nevertheless exhibited substantial currents at pH 7.4 that were pH sensitive, and traces that resembled those seen for ACD-5 or ACD-1. UNC-105-expressing oocytes exhibited very small currents, but these currents exhibited pH sensitivity, with a significant difference

between pH 4 and pH 7.4 (**$P = 0.0044$, 'Statistical summary document'). They thus merited more detailed characterisation. Interestingly, we also found three members, ASIC-1, ACD-2 and DEL-9, that robustly displayed increased currents at pH 4, i.e. that were opened, or further opened, in response to increasing proton concentrations (Fig. 3*C*; 'Statistical summary document'). We designated the members in this second group as putative acid-activated channels, which merited further characterisation.

The remaining family members (UNC-8c, ACD-4, DEG-1, DEGT-1, DELM-2, DEL-1, DEL2a, DEL2b, DEL-3, DEL-6, DEL-7, DEL-8b, DEL-10, MEC-4, MEC-10, EGAS-12, EGAS-2, EGAS-3 and EGAS-4) were tested in the same way but failed to exhibit currents

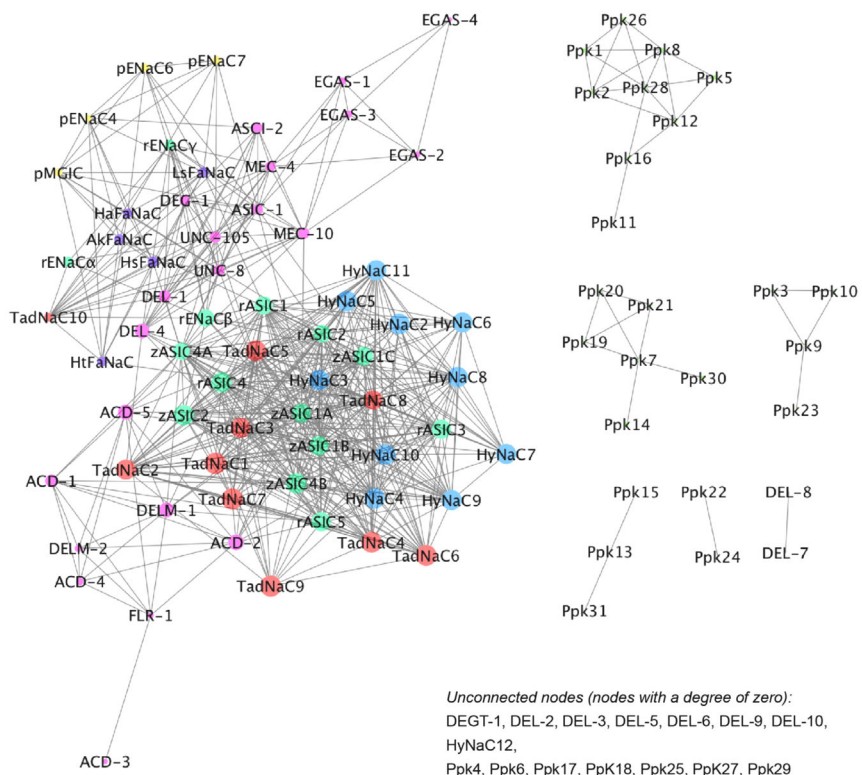

**Figure 2. Sequence similarity network (SSN) of the DEG/ENaC family**
SSN of diverse members of the DEG/ENaC family. Generated using the web tool for SSNs for protein families (EFI-EST) developed by the Enzyme Function Initiative efi.igb.illinois.edu/ (Gerlt et al., 2015; Gerlt, 2017; Zallot et al., 2018; Zallot et al., 2019). Each symbol represents a protein (node), two nodes are connected by a line (edge) if they share > 25% sequence similarity and lengths of edges correlate with the relative dissimilarities of each pair. Relative positioning of disconnected clusters and nodes has no meaning. Unconnected notes, i.e. nodes with a degree of zero, are indicated on the bottom right-hand side. Cytoscape (Shannon et al., 2003) was used to explore SSNs. Node sizes are determined by the degree of connectivity of the nodes (number of edges). The EFI-EST webtools use NCBI BLAST and CD-HIT to create SSNs. The computationally guided functional profiling tool uses the CGFP programs from the Balskus Lab (https://www.microbialchemist.com/metagenomic-profiling/) (Levin et al., 2017) and ShortBRED from the Huttenhower Lab (http://huttenhower.sph.harvard.edu/shortbred) (Kaminski et al., 2015). The data used to build these networks originated from the UniProt Consortium databases and the InterPro and ENA databases from EMBL-EBI. Node colouring is according to phyla, as indicated. Accession numbers can be found in Table 2. [Colour figure can be viewed at wileyonlinelibrary.com]

**Table 2. DEG/ENaC accession numbers**

| Organism | Protein name (accession number) |
| --- | --- |
| *Caenorhabditis elegans* | FLR-1 (UniProtKB/TrEMBL ID G5EGI5); ACD-1 (P91102); ACD-2 P91100); ACD-3 (G3MU02); ACD-4 (Q22970); ACD-5 (O01664); DELM-1 (O45402); DELM-2 (P91103); ASIC-1 (K7H9J0); ASIC-2 (Q22851); MEC-4 (P24612); DEL-4 (P91835); UNC-105 (Q09274); DEG-1 (P24585); DEL-1 (Q19038); MEC-10 (P34886); EGAS-1 (Q9U1T9); EGAS-2 (Q9U1T8); EGAS-3 (Q9XTS9); EGAS-4 (Q20852); DEGT-1 (Q19777); DEL-2 (G5ECD8); DEL-3 (Q93597); DEL-5 (G5EFH3); DEL-6 (Q8MPW0); DEL-7 (Q18651); DEL-8 (Q93205); UNC-8 (Q21974); DEL-9 (Q18077); DEL-10 (Q10025). |
| *Drosophila melanogaster* | PPK1 (Q7KT94); PPK2 (O46342); PPK3 (Q8MLR6); PPK4 (O61365); PPK5 (Q7KTW2); PPK6 (Q86LH3); PPK7 (Q9VME9); PPK8 (B7Z123); PPK9 (Q9W2B5); PPK10 (Q86LH1); PPK11 (Q9VL84); PPK12 (Q9W250); PPK13 (Q86LG9); PPK14 (Q86LG8); PPK15 (Q9VBF6); PPK16 (Q86LG7); PPK17 (Q9VJI4); PPK18 (Q9VL88); PPK19 (Q9VAJ3); PPK20 (Q86LG5); PPK21 (Q86LG4); PPK22 (Q8IMV2); PPK23 (Q9VX46); PPK24 (Q9V9Y5); PPK25 (A1Z6S4); PPK26 (Q9VS73); PPK27 (Q9VZN1); PPK28 (Q86LG1); PPK29 (A8DYP2); PPK30 (Q9VAJ5); PPK31 (A8JPJ8) |
| *Rattus norvegicus* | rASIC1 (P55926); rASIC2 (Q62962); rASIC3 (O35240); rASIC4 (Q9JHS6); rASIC5 (Q9R0W5); ENaCα (P37089); ENaCβ (P37090); ENaCγ (P37091) |
| *Danio rerio* | zAISC1A (Q708S7); zAISC1B (Q708S8); zAISC1C (Q708S6); zAISC2 (Q708S5); zAISC4A (Q708S4); zAISC4B (Q708S3) |
| *Hydra vulgaris* | HyNaC2 (A8DZR6); HyNaC3 (A8DZR7); HyNaC4 (A8DZR8); HyNaC5 (D3UD58); HyNaC6 (A0A0A0MP54); HyNaC7 (A0A0A0MP73); HyNaC8 (A0A0A0MP55); HyNaC9 (A0A0A0MP48); HyNaC10 (A0A0A0MP61); HyNaC11 (A0A0A0MP67); HyNaC12 (A0A0A0MP74) |
| *Aplysia kurodai* | FaNaC (Q4H3×6) |
| *Platynereis dumerilii* | pENaC4 (A0A2S1B6I2); pENaC7 (A0A2S1B6Q3); pENaC6 (A0A2S1B6R1); pMGIC (A0A2S1B6I3) |
| *Lottia gigantean* | LgFaNaC (V4C2H5) |
| *Lymnaea stagnalis* | LsFaNaC (Q9BJD0) |
| *Helix aspersa* | HaFaNaC (Q25011). |
| *Helisoma trivolvis* | HtFaNaC (Q9NBC7) |
| *Trichoplax adhaerens* | TadNaC1 (B3S0Z3); TadNaC2 (A0A5J6BSS6); TadNaC3 (A0A5J6BTF2); TadNaC4 (A0A5J6BSQ9); TadNaC5 (A0A5J6BSM6); TadNaC6 (A0A5J6BVG3); TadNaC7 (A0A5J6BSR6); TadNaC8 (A0A5J6BV03); TadNaC9 (A0A5J6BWR3); TadNaC10 (A0A5J6BSU1). |

that were significantly different from the water-injected controls ('Statistical summary document'), precluding any conclusion about their pH sensitivity. This may be because they are not properly trafficked to the cell surface, due to dependence on a missing factor or subunit, and/or because they do not form functional channels as homomers. ASIC2b, for example, is non-functional as a homomer but modulates the properties of other subunits when co-expressed (Lingueglia et al., 1997); likewise, *C. elegans* FLR-1, ACD-3 and DEL-5 exhibit acid-sensitive currents only as heteromers (Kaulich et al., 2022).

## *C. elegans* acid-sensitive DEG/ENaCs vary in their pH dependence and degree of ion permeability

To characterise the acid sensitivity and ion selectivity of the acid-sensitive DEG/ENaCs, we perfused different pH solutions over the oocytes expressing the respective cRNA and substituted different monovalent cations in the recording solution. The ion substitution experiments were performed at $pH_{50}$ for acid-activated channels and at pH 7.5 for acid-inhibited channels and the shift in reversal potential ($\Delta E_{rev}$) was assessed after replacing NaCl with either KCl or LiCl in the solution containing 96 mM NaCl, 1 mM MgCl$_2$ and 5mM MES, based on a previous protocol (Hardege et al., 2015). The osmolarity was adjusted using D-glucose. All ion-substitution experiments were conducted in the absence of Ca$^{2+}$ as it has been shown to block ASICs (Paukert et al., 2004). We first investigated the properties of the acid-inhibited channels. We and others previously reported that ACD-5 is permeable to Li$^+$, and to a lesser degree to Na$^+$ and K$^+$, but not Ca$^{2+}$, and is inhibited by both low and high pH, with $pH_{50}$ values of 4.87 and 6.48 (Kaulich et al., 2022), whereas ACD-1 is Na$^+$-selective, with a $pH_{50}$ of 6.4 (Wang et al., 2008). We found that DELM-1 was inhibited by acidic pH, with a $pH_{50}$ of 5.1 (in agreement with previous evidence that it is constitutively open at neutral pH; Han et al., 2013). Co-expression with DELM-2 (with which it is co-expressed in glia; Han et al., 2013) did not change the $pH_{50}$ (Fig. 4*A* and *B*). This is in line with previous research

which did not find any significant difference in current amplitude, amiloride sensitivity or ion selectivity when co-expressing these subunits together (Han et al., 2013). Either of the genes can rescue phenotypes arising from single mutants, and they therefore might be redundant or function with another subunit not yet identified. However, co-expression experiments in oocytes can be difficult to interpret, due to the potential presence of a mix of homo- and heteromeric channels, which can confound whole-cell recordings (Hesselager et al., 2004), and we cannot exclude the possibility that DELM-2 is either not expressed or not translocated.

The $Na^+$- and $K^+$-permeable UNC-105 has previously been implicated in proton-sensing *in vivo* in muscles (Jospin & Allard, 2004). We found that UNC-105 (isoform h) produced small currents which were inhibited by increasing proton concentrations, in the range of pH 9–4, with a $pH_{50}$ of 6.30 (Fig. 4*A* and *B*). Our ion substitution experiments showed a median negative shift in $E_{rev}$ of 15 mV when shifting from a NaCl solution to a KCl solution, a median positive shift in $E_{rev}$ of 9.5 mV when shifting from a NaCl solution to a LiCl solution, and a median negative shift in $E_{rev}$ of 24 mV when shifting from the basal sodium buffer ND96 to a $Na^+$ replacement buffer with impermeant *N*-methyl-D-glucamine buffer at pH 7.4 (Fig. 4*C* and *D*). Permeability ratios for each of the channels were calculated as previously described (Lynagh et al., 2020; see Methods), giving a $P_{Na}/P_{Li}$ under 1 and

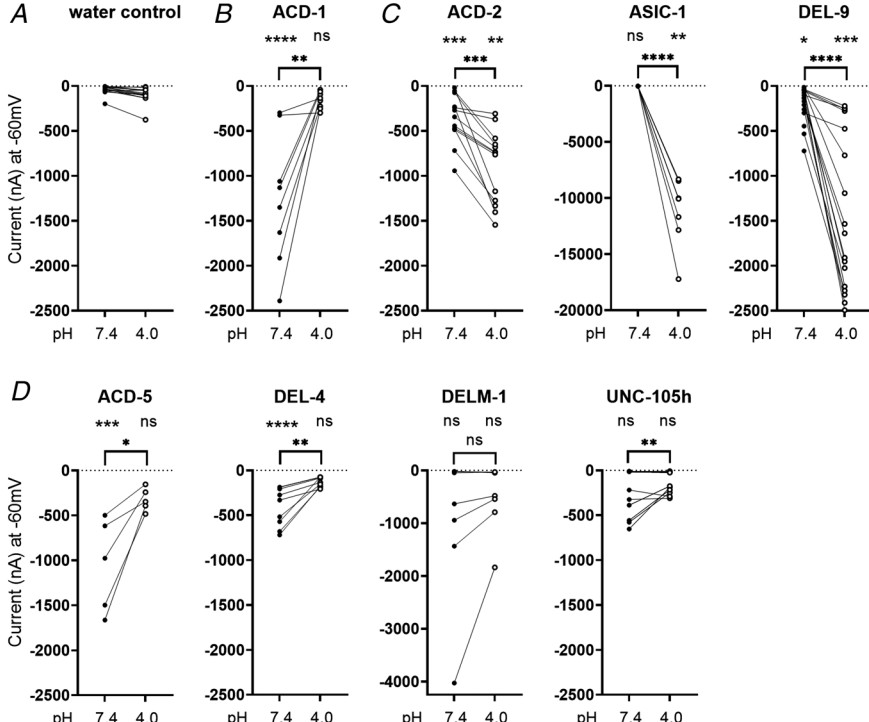

**Figure 3. Quantification of current at pH 7.4 and pH 4 of *Xenopus* oocytes expressing *C. elegans* DEG/ENaC subunits**

Graphs show raw current upon perfusion with either pH 7.4 (filled circle) or pH 4 (open circle) as indicated, for *Xenopus* oocytes injected with the respective cRNA. Lines connect data from individual oocytes. *A*, nuclease free water-injected oocytes serve as negative controls ($N = 13$). *B*, ACD-1 ($n = 8$) expressing oocytes serve as a positive control (Wang et al., 2008). *C*, constructs that form acid-activated channels: ACD-2 ($N = 13$), ASIC-1 ($N = 7$) and DEL-9 ($N = 19$). *D*, constructs that form acid-inhibited channels: ACD-5 ($N = 5$), DEL-4 ($N = 9$), DELM-1 ($N = 7$) and UNC-105 ($N = 17$). The pH-insensitive currents of FLR-1, ACD-3 and DEL-5 are published elsewhere (Kaulich et al., 2022). For the remaining 19 DEG/ENaC subunits that did not display acid-sensitive currents, see 'Statistical summary document'. Holding potential was −60 mV. Experiments were repeated on more than three different days, with different oocyte batches, and were pooled together. Note that all graphs are plotted using the same scale, except ASIC-1 and DELM-1 due to the large currents observed in oocytes expressing these two constructs. Results of statistical tests are shown above graphs. Top row: comparison with water-injected control, at pH 7.4 and 4.0, respectively; bottom row: comparison between pH 7.4 and pH 4.0. *P*-values, listed in the order shown, left to right, top to bottom, for each subunit: ACD-1 (<0.0001, >0.9999, 0.0037), ACD-2 (0.0004, 0.0037, 0.0002), ASIC-1 (>0.9999, 0.0051, <0.0001), DEL-9 (0.0102, 0.0005, <0.0001), ACD-5 (0.0004, 0.6178, 0.0193), DEL-4 (<0.0001, >0.9999, 0.0025) DELM-1 (0.2448, >0.9999, 0.1608), UNC-105 (0.3270, >0.9999, 0.0044). Asterisks indicate significance level, **** <0.0001, *** <0.001, ** <0.01, * <0.05.

**Table 3. Permeability ratios for UNC-105h, ACD-2, ASIC-1 and DEL-9**

| | | $P_{Na}/P_K$ (mean and SD) | $P_{Na}/P_{Li}$ (mean and SD) | Selectivity sequence |
|---|---|---|---|---|
| Acid-inhibited | UNC-105h | 1.84 (0.28) | 0.68 (0.07) | $Li^+ > Na^+ = K^+$ |
| Acid-activated | ACD-2 | 4.49 (2.28) | 1.11 (0.70) | $Na^+ > Li^+ > K^+$ |
| | ASIC-1 | 0.62 (0.04) | 0.69 (0.08) | $K^+ = Li^+ > Na^+$ |
| | DEL-9 | 1.02 (0.12) | 0.98 (0.30) | $Na^+ = Li^+ = K^+$ |

$P_{Na}/P_K$ over 1 (Table 3). This is in line with evidence from a gain-of-function mutant form of UNC-105 and characterisation of currents in muscle (Garcia-Anoveros et al., 1998; Jospin & Allard, 2004). DEL-4 was shown to be $Na^+$-selective and blocked by increasing proton concentrations, in the range pH 8–4, with $pH_{50}$ of 5.7 (D. Petratou, M. Gjikolaj, E. Kaulich, W. R. Schafer, N. Tavernarakis, unpublished observations).

By contrast, for all three acid-activated subunits, acid-evoked currents increased in a

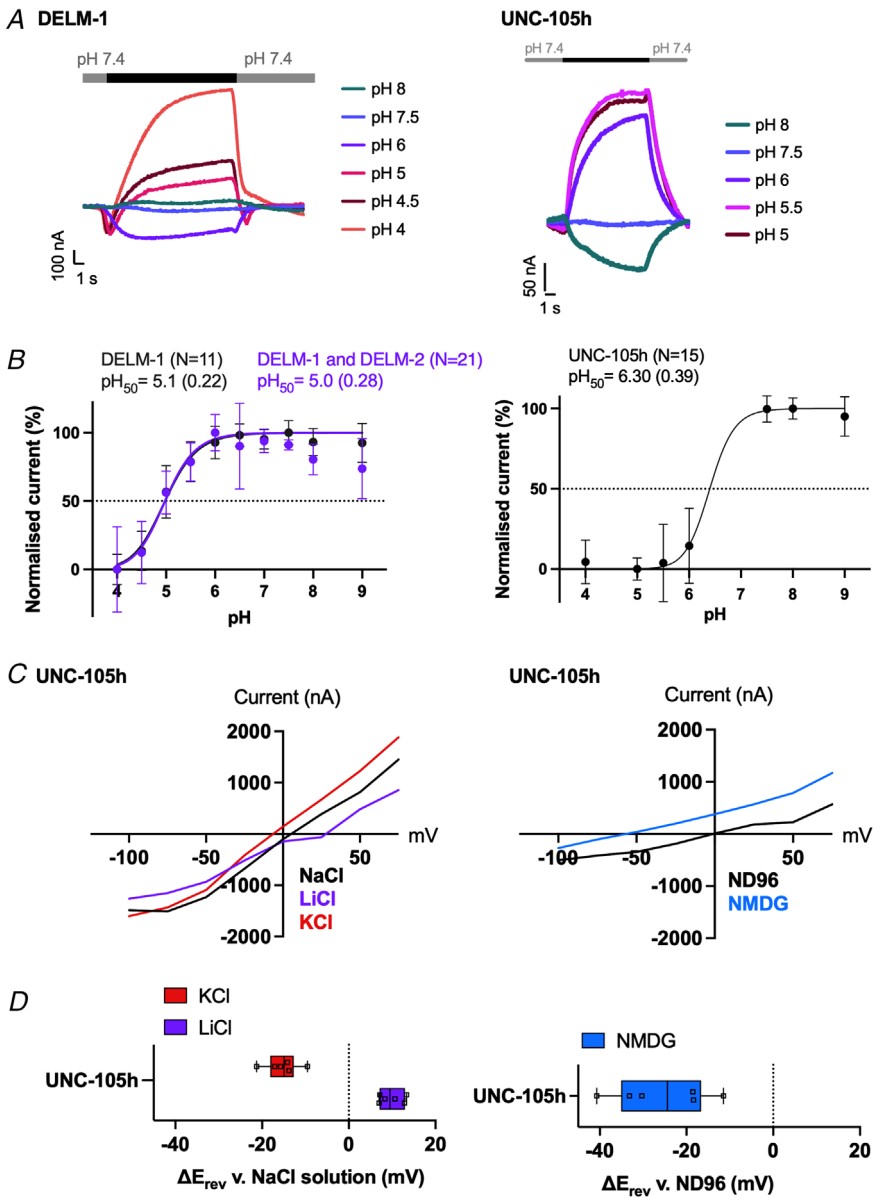

**Figure 4. DELM-1 and UNC-105 form acid-inhibited homomeric cation channels**

*A*, representative traces of *Xenopus* oocytes expressing the respective construct when perfused with solutions of different pH (black bar) from a pre-stimulus holding pH of 7.4. *B*, current–pH relationship. Dotted line indicates $pH_{50}$. DELM-1, $pH_{50} = 5.1$ (SD = 0.22, $N = 11$); and DELM-1 co-expressed with DELM-2, $pH_{50} = 5.0$ (SD = 0.28, $N = 21$), Global comparison of fits (extra sum-of-squares *F*-test) showed that the $pH_{50}$ values of both datasets are the same $P = 0.93$. UNC-105 isoform h, $pH_{50} = 6.30$ (SD = 0.39, $N = 15$). Currents were recorded at a holding potential of –60 mV, baseline subtracted using pre-stimulus current and drift corrected, then normalised to maximal/maximal currents and best fitted with the Hill equation (variable slope). Data points and error bars represent mean (SD). *C*, average calculated from $N = 6$ oocytes of $\Delta E_{rev}$ when shifting from a NaCl solution to KCl or LiCl solution. A negative shift of $E_{rev}$ indicates a preference for $Na^+$ over the respective ion and a positive shift indicates a preference of the respective ion over $Na^+$. Data are presented as box-plots (hinges of the plot are the 25th and 75th percentiles) with median and whiskers of minimum and maximum. *D*, representative current–voltage (*I–V*) relationships for oocytes expressing UNC-105h. Currents are baseline subtracted with Roboocyte2+ software. The oocyte membrane was clamped at –60 mV and voltage steps from –150 to +75 mV were applied as indicated. Currents in μA (*y*-axis), voltage steps in mV (*x*-axis) as indicated. [Colour figure can be viewed at wileyonlinelibrary.com]

concentration-dependent manner. We observed $pH_{50}$ values of 4.50 for ASIC-1, 5.04 for ACD-2 and 4.33 for DEL-9. (Fig. 5*A* and *B*). None of the three homomeric channels desensitised, but reached a plateau after several seconds, similar to what has been observed for

the lamprey ASIC1 (Li, Yang et al., 2011), with ASIC-1 and ACD-2 reaching the plateau much faster than DEL-9, as shown by the time scale in Fig. 5*A* and *B*. This is not an artefact of the perfusion rate as the same perfusion rate was used for all oocytes and conditions. As no

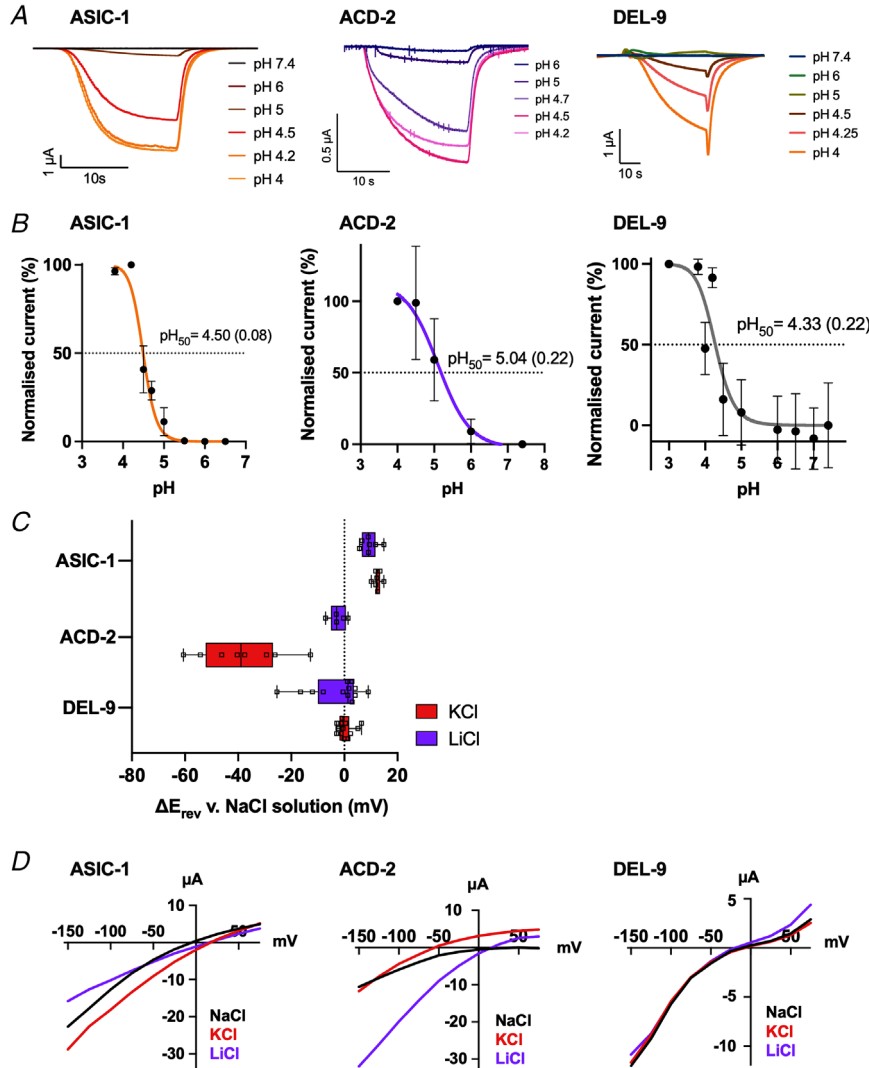

**Figure 5. ASIC-1, ACD-2 and DEL-9 form acid-activated homomeric channels permeable to monovalent cations**

*A*, representative traces of *Xenopus* oocytes expressing the respective construct when perfused with solutions of different pH, lowering the pH from a holding pH of 7.4. *B*, current–pH relationship. Dotted line indicates $pH_{50}$. ASIC-1, $pH_{50}$ = 4.50 (SD = 0.08, *N* = 13); ACD-2, $pH_{50}$ = 5.04 (SD = 0.22, *N* = 6); DEL-9, $pH_{50}$ = 4.33 ± 0.22 (*N* = 10). Currents were recorded at a holding potential of −60 mV; baseline subtracted using pre-stimulus current and drift corrected, then normalised to maximal currents ($I/I_{max}$) and best fitted with the Hill equation (variable slope). Data points and error bars represent mean (SD). *C* and *D*, summary of ion selectivity. ASIC-1, ACD-2 and DEL-9 are permeable to monovalent cations. *C*, average calculated from 4 < *N* < 12 oocytes for each construct of $\Delta E_{rev}$ when shifting from a NaCl solution to KCl or LiCl solution. A negative shift of $E_{rev}$ indicates a preference for $Na^+$ over the respective ion and a positive shift indicates a preference of the respective ion over $Na^+$. Data are presented as box-plots (hinges of the plot are the 25th and 75th percentiles) with median and whiskers of minimum and maximum. *D*, representative current–voltage (*I–V*) relationships for oocytes expressing ASIC-1, ACD-2 and DEL-9. Currents are baseline subtracted with Roboocyte2+ software. For acid-evoked current at $pH_{50}$ concentrations, see panel *B* for reference. The oocyte membrane was clamped at −60 mV and voltage steps from −150 to +75 mV were applied as indicated. Currents in μA (*y*-axis), voltage steps in mV (*x*-axis) as indicated. [Colour figure can be viewed at wileyonlinelibrary.com]

physiological data were available for any of the three acid-activated DEG/ENaC subunits, we investigated their ion selectivity by carrying out ion substitution experiments (Fig. 5*C* and *D*). Permeability ratios for each of the channels were calculated as previously described (Lynagh et al., 2020; see Methods) and are shown in Table 3. For ASIC-1 expressing oocytes, our results showed a median positive shift in $E_{rev}$ of 12.2 mV, when shifting from a NaCl solution to a KCl or LiCl solution ($P_{Na}/P_{Li}$ and $P_{Na}/P_K$ under 1). In contrast ACD-2 was observed to be a sodium channel, selective for Li$^+$ and Na$^+$ over K$^+$ with no change in $E_{rev}$ when switching from a NaCl to a LiCl solution, but a large median negative shift in $E_{rev}$ of 43.3 mV when shifting from a NaCl solution to a KCl solution ($P_{Na}/P_{Li}$ around 1 and $P_{Na}/P_K$ over 4). This is similar to that previously described for DELM-1 (Han et al., 2013). Finally, DEL-9 was non-selective for monovalent cations, showing no shift in reversal potential when switching between solutions ($P_{Na}/P_{Li}$ and $P_{Na}/P_K$ of approximately 1), similar to the previously described heteromeric cnidarian HyNaC (Dürrnagel et al., 2012). Thus, the newly characterised acid-sensitive DEG/ENaCs showed significant diversity in their ion permeability properties, consistent with previous findings on other family members (Canessa et al., 1994; Carattino & Della Vecchia, 2012; Fechner et al., 2021; Wang et al., 2008).

### Contribution of conserved 'GAS' belt on functionality of acid-activated and acid-inhibited *C. elegans* DEG/ENaCs

To verify whether the acid-evoked currents of DEL-9-expressing oocytes and the small currents observed at neutral pH in UNC-105h-expressing oocytes are due to channel activity and not due to endogenous currents, we performed two experiments. The first was to demonstrate a concentration-depended increase in current as a result of increased channel expression on the surface of the oocyte. For this experiment, we injected the constructs at 250, 500 and 750 ng/μl, and compared to the acid-evoked currents of nuclease water-injected control oocytes (0 ng/μl) at a fixed acidic pH. Results showed a concentration-dependent increase in median acid-evoked current for DEL-9 expressing oocytes (Fig. 6*A*). Similarly, we saw a concentration-dependent current increase at neutral pH for UNC-105h-expressing oocytes (Fig. 6*B*). The enhancement of currents dependent on the amount of cRNA injected demonstrates that the currents observed are due to channel activity.

The second experiment to show that the observed currents were not endogenous but due to channel activation was to render the channel non-functional. We chose the deletion of three conserved residues in the second transmembrane domain, the 'Gly–Ala–Ser'

(GAS) belt which corresponds to the residues Gly443–Ala444–Ser445 in the chicken cASIC1. The 'GAS' belt is highly conserved across all members of the DEG/ENaC/ASIC family and mutation studies have shown that these residues are important for ion selectivity and forming a functional stable pore (Baconguis et al., 2014; Carattino & Della Vecchia, 2012; Chen et al., 2022; Kellenberger Gautschi et al., 1999; Li, Yang et al., 2011; Lynagh et al., 2017, 2020). Although the 'GAS' belt is not involved directly in the ion selectivity, the full-length structure of cASIC1a showed that it interacts with another conserved motif (HG-motif) to stabilise the channel pore, thus indirectly contributing towards the ion permeation pathway (Yoder & Gouaux, 2020). Based on these findings, we deleted the 'GAS' belt residues, which we hypothesised would render the channel non-functional. The 'GAS' belt deletions abolished both DEL-9 acid-evoked currents and UNC-105h pH 7.4-evoked currents (Fig. 6*C–F*). These results show that the currents are indeed due to the channels' activity and fit well with previous results mentioned above showing that the conserved 'GAS' belt is important for maintaining a functional channel.

### Exploring the effect of known DEG/ENaC modulators, amiloride and zinc

The anti-hypertensive amiloride is a potent blocker of many, but not all, DEG/ENaCs; homomeric ASIC3, ASIC2 and heteromeric ASIC3/ASIC1b are potentiated by amiloride, whereas *C. elegans* DEGT-1 is amiloride-insensitive (Adams et al., 1999; Baconguis et al., 2014; Bentley, 1968; Besson et al., 2017; Canessa et al., 1994; Fechner et al., 2021; Kellenberger et al., 2003; Li, Yu et al., 2011; Matasic et al., 2021; Palmer & Frindt, 1986; Schild et al., 1997). Therefore, we investigated if the acid-evoked currents at the respective pH$_{50}$ of ASIC-1, ACD-2 and DEL-9 could be modulated by amiloride (Fig. 7; for pH$_{50}$ see Fig. 5*B* for reference). We observed that ASIC-1 and ACD-2 are sensitive to amiloride; indeed acid-evoked currents could be blocked in a dose-dependent manner, which is also a common characteristic of the DEG/ENaC superfamily (Vullo & Kellenberger, 2020). Amiloride blocked ASIC-1 acid-evoked currents with an IC$_{50}$ of 108 μM (Fig. 7*B*) and ACD-2 acid-evoked currents with an IC$_{50}$ of 87 μM (Fig. 7*C*) (note that Fig. 7 shows currents that are baseline-subtracted using the pre-stimulus current, i.e. at pH 7.4, at which the channel is maximally inhibited by pH; Fig. 5*A* and *B*). ACD-2 leak currents at pH 7.4 are also blocked by amiloride (Fig. 8*B*), supporting the idea that the channel is slightly open at this pH. In contrast, DEL-9 acid-activated currents at pH 4 were insensitive to amiloride (Fig. 7*D*). Interestingly, DEL-9 expressing oocytes showed a leak current at pH

7.4 which was slightly potentiated in the presence of amiloride (Fig. 8*C*), a phenomenon that has previously been described for another *C. elegans* DEG/ENaC, DEL-5 (Kaulich et al., 2022), and the mammalian ASIC3, possibly wedging the pore open with the binding of multiple amiloride molecules (Adams et al., 1999; Matasic et al., 2021).

We also investigated the effect of amiloride on the acid-inhibited subunits, showing that ACD-1, DELM-1

and UNC-105 and ACD-5 are all inhibited (Fig. 8*A* and *B*), in agreement with previous observations (Han et al., 2013; Jospin & Allard, 2004; Kaulich et al., 2022; Wang et al., 2008). Likewise, DEL-4 is inhibited by amiloride (D. Petratou, M. Gjikolaj, E. Kaulich, W. R. Schafer, N. Tavernarakis, unpublished observations). UNC-105 showed paradoxical modulation by amiloride, with an $EC_{50}$ of 6.7 μM amiloride and a strong inhibition at 1 μM (Fig. 9*A* and *B*).

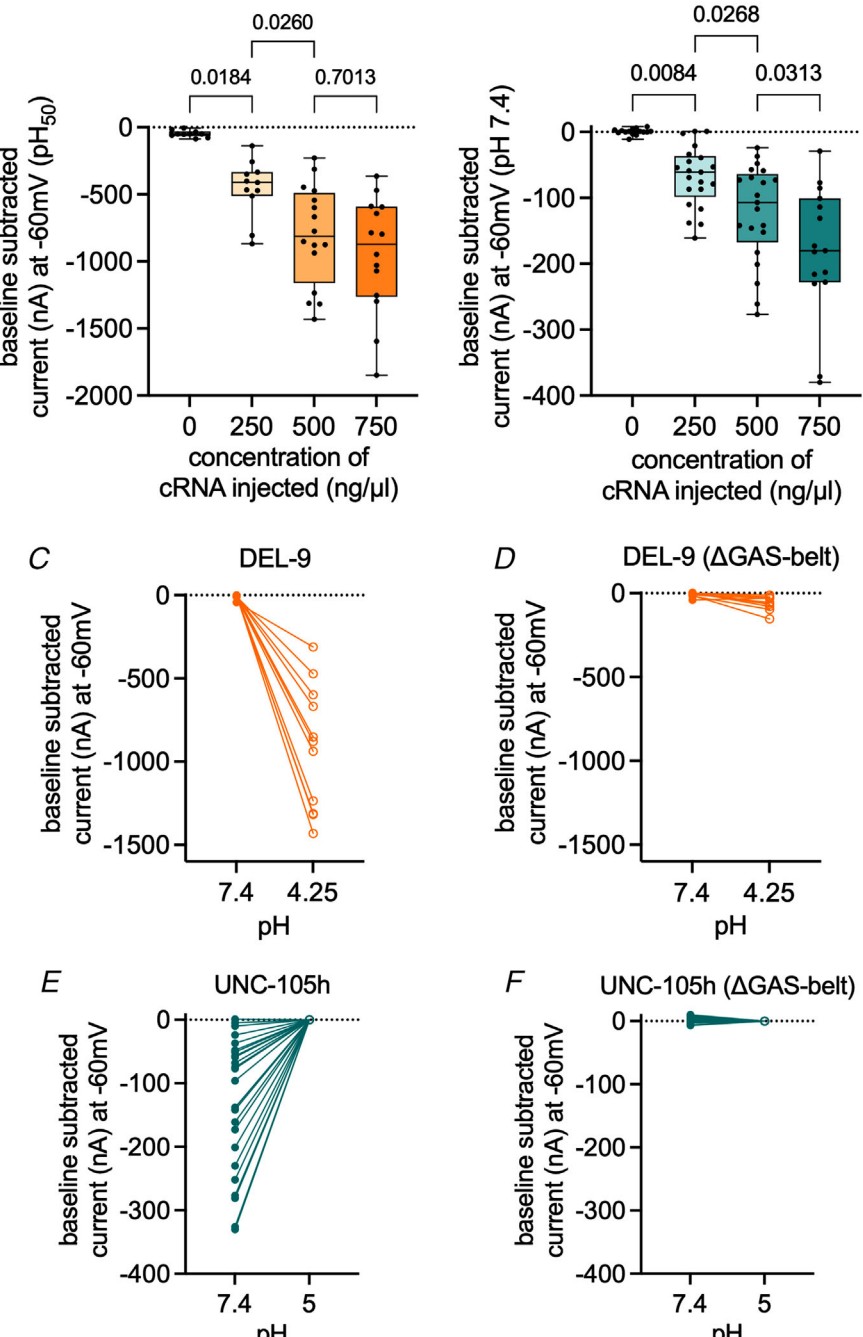

**Figure 6. Concentration-dependent acid-sensitive currents and ΔGAS belt deletion of UNC-105h and DEL-9 channels**

*A* and *B*, baseline-subtracted peak currents of oocytes expressing DEL-9 (*A*) and UNC-105 (*B*) at varying concentrations (0 ng/μl, nuclease-water injected controls), recorded at $pH_{50}$ concentration for DEL-9 (see Fig. 4*B* for $pH_{50}$) and at pH 7.4 for UNC-105h). Number of oocytes: $N_{DEL-9}$ (left to right) = 10, 11, 16, 14; and $N_{UNC-105h}$ (left to right) = 15, 21, 21, 15. A one-way-ANOVA with *post hoc* test (Bonferroni correction) was conducted between the different concentrations, indicated are the *P*-values. Currents in nA (*y*-axis), concentration of injected cRNA in ng/μl (*x*-axis) as indicated. *C* and *D*, baseline subtracted currents of DEL-9 wild-type (*N* = 11 oocytes) and DEL-9 (ΔGAS-belt) (*N* = 10 oocytes) (injected at 500 ng/μl) at pH 7.4 and pH 4.25. *E* and *F*, baseline subtracted peak currents of UNC-105h wild-type (*N* = 20 oocytes) and UNC-105h (ΔGAS-belt) (*N* = 29 oocytes) (injected at 500 ng/μl) at pH 7.4 and pH 5. The oocyte membrane was clamped at −60 mV. Currents in nA (*y*-axis), pH (*x*-axis) as indicated. For DEL-9, currents were baseline subtracted using pre-stimulus current (i.e. at pH 7.4), then drift corrected, using Roboocyte2+ software; for UNC-105, this was baseline adjusted using the pH 5 current (since current is minimal at pH 5; Fig. 4*B*). [Colour figure can be viewed at wileyonlinelibrary.com]

Zinc is an essential trace element in the brain that is present in synaptic vesicles and co-released with neurotransmitters (Blakemore & Trombley, 2017; Takeda et al., 2004). High concentrations (in the range of 1 μM to 10 mM) potentiate acid-evoked currents of homomeric and heteromeric ASIC2a-containing channels (Baron et al., 2001). By contrast, $Zn^{2+}$ can block ASIC1b subunits, via multiple proposed binding sites in the extracellular domain (Baron et al., 2001; Jiang et al., 2012). We therefore investigated the effect of $Zn^{2+}$ on the newly characterised acid-sensing channels. We found that the $pH_{50}$-evoked transient currents of the acid-activated channels could be blocked by $Zn^{2+}$ in a dose-dependent manner, with $IC_{50}$ values of 284 μM for ASIC-1, 51 μM for ACD-2

and 23 μM for DEL-9 (Fig. 10). We further tested $Zn^{2+}$ modulation at pH 7.4 (which corresponds to the open state of the channels) of the acid-inhibited channels and found that, similar to the acid-activated channels, the acid-inhibited ACD-1, DEL-4 and UNC-105 were blocked by increasing concentrations of $Zn^{2+}$ with $IC_{50}$ values of 208, 12 and 31 μM, respectively (Fig. 11A, D and E). However, $Zn^{2+}$ had a dual effect on ACD-1 depending on the concentration, with low concentrations (up to 10 μM) being slightly current enhancing and concentrations above 10 μM strongly inhibiting. ACD-5 also showed a 'dual' response with an $IC_{50}$ of 190 nM in the range of 0.1–100 μM (likely to reflect saturation of the channel) and an $EC_{50}$ of 1251 μM in the range of 100–5000 μM

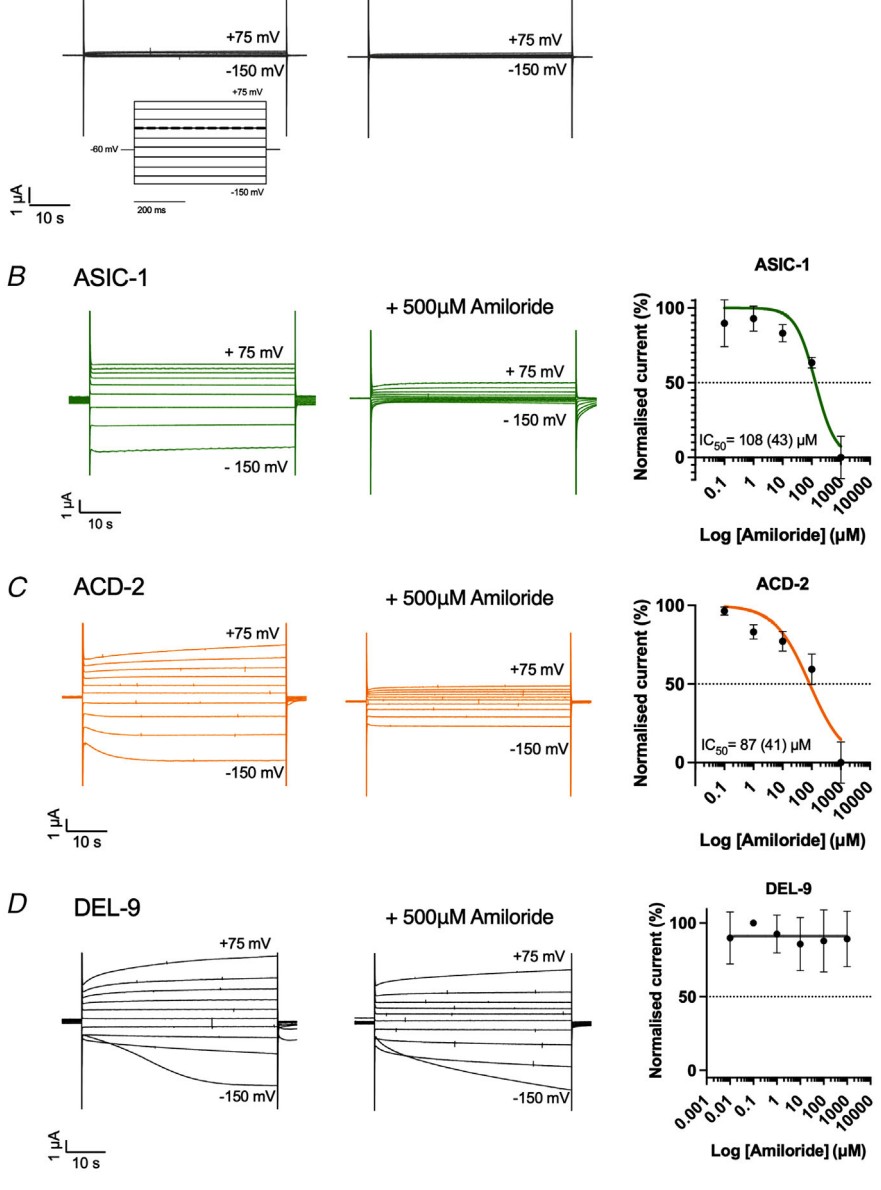

**Figure 7. Amiloride-sensitivity of acid-activated channels ACD-2, ASIC-1 and DEL-9 at $pH_{50}$**
ASIC-1 and ACD-2 (but not DEL-9) acid-evoked transient currents can be blocked by amiloride. Representative acid-evoked transient currents in the absence (left) and presence of 500 μM amiloride (middle) at $pH_{50}$ (for $pH_{50}$ of each channel see Fig. 5B), and amiloride dose–response curves (right). A, nuclease-free water-injected oocytes (negative control) are unaffected by low pH (example shown here pH 4.5) or amiloride. B and C, amiloride dose response of ASIC-1 with an $IC_{50}$ of 108 μM amiloride (SD = 43, N = 10) (B) and ACD-2 with an $IC_{50}$ of 87 μM amiloride (SD = 41, N = 12) (C). D, DEL-9 (N = 3) acid-activated transient currents are unaffected by amiloride. The *Xenopus* oocyte membrane was clamped at −60 mV and voltage steps from −150 to +75 mV were applied as indicated. Raw currents in μA (y-axis), voltage steps in mV (x-axis) as indicated. Data points and error bars represent mean (SD). Currents were recorded at a holding potential of −60 mV. For dose responses, currents were baseline subtracted (i.e. zero indicates the pre-stimulus (pH 7.4) current), then normalised to the maximum current ($I_{max}$) calculated for each oocyte individually, and best fitted with the Hill equation (variable slope). [Colour figure can be viewed at wileyonlinelibrary.com]

(Fig. 11*B*). Similar bidirectional effects have been previously described for the heteromeric ASIC1a/3 which is potentiated by $Zn^{2+}$ at concentrations up to 100 μM, but potentiated at $Zn^{2+}$ concentrations over 250 μM (Jiang et al., 2020). In contrast, increasing concentrations of $Zn^{2+}$ potentiated the baseline currents of DELM-1, with an $EC_{50}$ of 263 μM (Fig. 11*C*). These findings highlight that, like their mammalian counterparts, these acid-sensitive DEG/ENaCs can be modulated by $Zn^{2+}$ in an inhibiting or potentiating manner. This demonstrates that *C. elegans* acid-sensitive DEG/ENaCs share the function of zinc and amiloride modulation with their vertebrate homologues.

## Acid-activated DEG/ENaCs are expressed in both neurons and muscles

To characterise the expression patterns of the acid-activated *C. elegans* DEG/ENaCs, we used transcriptional reporters, fusing a fluorophore gene downstream of the appropriate promoter sequences. We confirmed previous reports that the *asic-1* promoter drives expression in the ADE, CEP, PVQ, PDE and PVD neurons (De Stasio et al., 2018; Husson et al., 2012; Voglis & Tavernarakis, 2008) and also observed expression in FLP and ventral cord neurons (Fig. 12). The *acd-2*

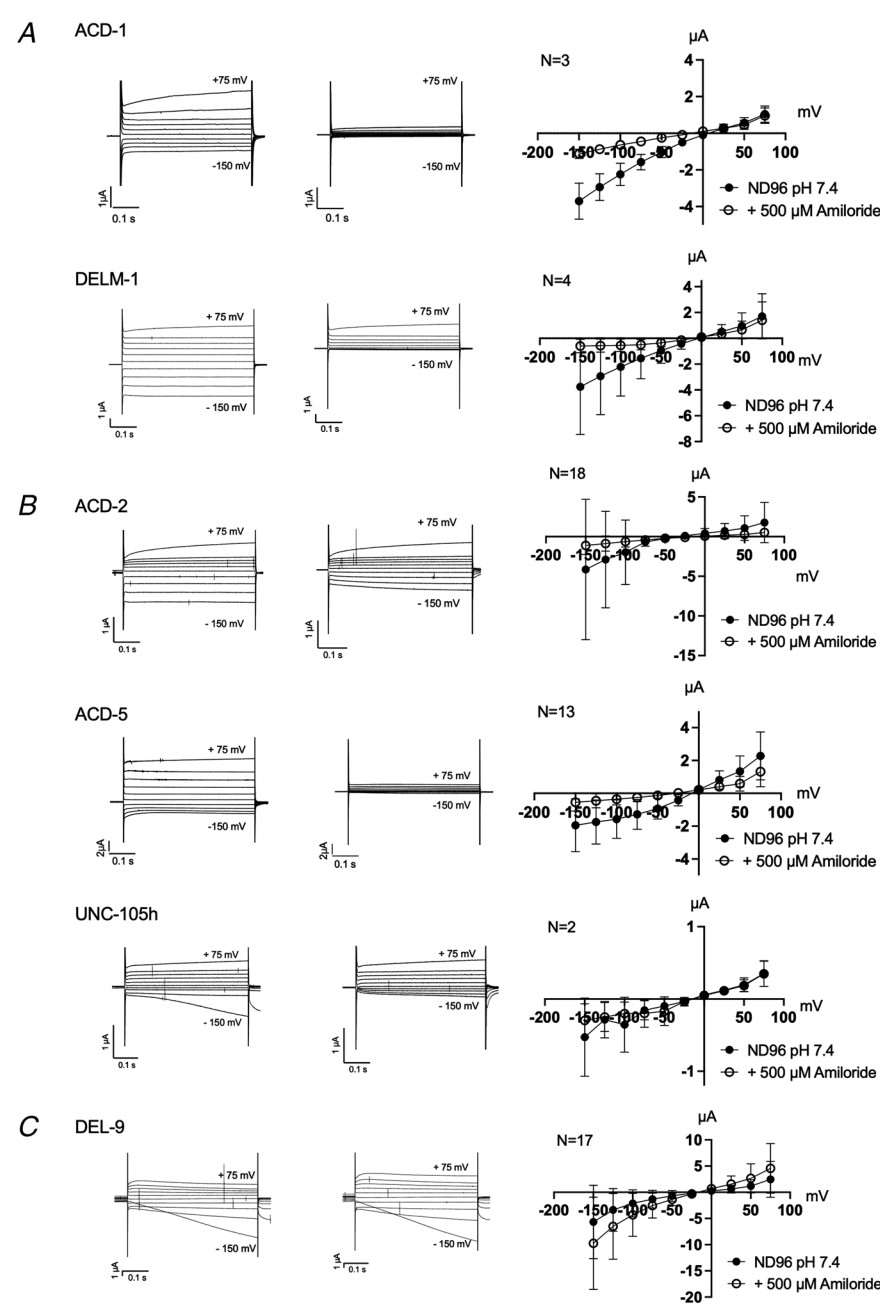

**Figure 8. The effect of amiloride on acid-sensitive DEG/ENaCs at pH 7.4**
Representative transient currents in the absence (left) and presence of 500 μM amiloride (middle), and current–voltage (*I–V*) relationships (right) at pH 7.4. *A*, ACD-1 and DELM-1 (positive controls). *B*, ACD-2, ACD-5, DEL-4, DEL-10 and UNC-105h transient currents can be blocked by amiloride. *C*, DEL-9 transient currents at pH 7.4 are potentiated in the presence of amiloride. *Xenopus* oocytes are perfused with a basal solution (ND96) only (filled circles), and in presence of the DEG/ENaC channel blocker amiloride (open circles). Scales vary between constructs due to the variability in the amplitude of the currents. The oocyte membrane was clamped at −60 mV and voltage steps from −150 to +75 mV were applied as indicated. Raw currents in μA (*y*-axis), voltage steps in mV (*x*-axis) as indicated. Data points and error bars represent mean and SD, and *N* is number of individual oocytes tested in independent experiments pooled together.

transcriptional reporter showed low-level expression in unidentified anterior neurons or glia in the head based on localisation and previous RNA-sequencing data (Cao et al., 2017). Our *del-9* reporter was expressed in anterior and posterior body wall muscles, egg-laying muscles as well as head and tail neurons including AVL and PVQ. Thus, our results, also in line with previous reports, show that the *C. elegans* acid-sensing DEG/ENaCs are not confined to a particular tissue but can be expressed in both neuronal and non-neuronal tissues.

## Discussion

The acid-sensing members of the DEG/ENaCs, the ASICs, are proton receptors that can sense changes in extracellular pH in neuronal and non-neuronal tissues. *C. elegans* is a genetic model system used to pioneer research into DEG/ENaCs, but relatively little is known about pH sensing by *C. elegans* DEG/ENaCs or their modulation by other molecules. We have shown here that there are at least three *C. elegans* acid-activated DEG/ENaCs, ASIC-1, ACD-2 and DEL-9. All three acid-activated channels are cation channels that are activated by increasing proton concentrations and inhibited by the trace element zinc. ASIC-1 and ACD-2 can also be blocked by the anti-hypertensive drug amiloride in a dose-dependent manner (see Table 4 for a summary of channel properties).

We also cannot exclude the possibility that *C. elegans* possesses additional acid-sensitive DEG/ENaCs. For those channels in our screen which did not exhibit significantly different currents from the water-injected controls, it is likely that some require additional subunits to form a functional channel, or additional factors for cell surface localisation or to enhance currents. Further characterisation of their function will thus require co-expression of possible partners for heteromer

formation, identified from shared expression patterns or phenotypes, and/or verification of trafficking to the cell surface.

Our findings raise the question of the channels' functional roles *in vivo*. In particular, it is unclear whether these channels would encounter a prolonged acidic environment under physiological conditions, especially as the acid-sensing DEG/ENaCs differ from the vertebrate ASICs in that they do not desensitise during the course of the proton stimulation (whereas the vertebrate ASICs do so within milliseconds). For instance, the murine ASIC1 is activated by protons released from synaptic vesicles during neurotransmission, which might constitute a short increase in acidification, but could also represent a highly variable acidic environment depending on the rate of exocytosis (Du et al., 2014). Protons are co-packed in presynaptic vesicles with other neurotransmitters by the action of the proton pump vacuolar-type ATPase (V-ATPase) (Gowrisankaran & Milosevic, 2020), and then co-released into the synaptic cleft, inducing a brief local drop in pH of 0.2–0.6 units (Du et al., 2014; Miesenbock et al., 1998; Zeng et al., 2015). This in turn stimulates postsynaptic receptors such as the ASICs (Soto et al., 2018). Similarly, presynaptic vesicles also co-release milli-molar concentrations of $Zn^{2+}$ during synaptic transmission (Assaf & Chung, 1984; Blakemore & Trombley, 2017; Frederickson & Moncrieff, 1994; Howell et al., 1984; Takeda et al., 2004), which could in turn modulate ASIC channels during neurotransmission.

A role in neurotransmission may also be relevant to the *in vivo* function of *C. elegans* channels, in particular ASIC-1. We have shown that *C. elegans* ASIC-1, like the murine ASIC1, can be activated by external protons in a concentration-dependent manner, suggesting that it might be involved in synaptic transmission in a similar way. This would fit well with behavioural and genetic evidence that ASIC-1 localises to presynaptic terminals

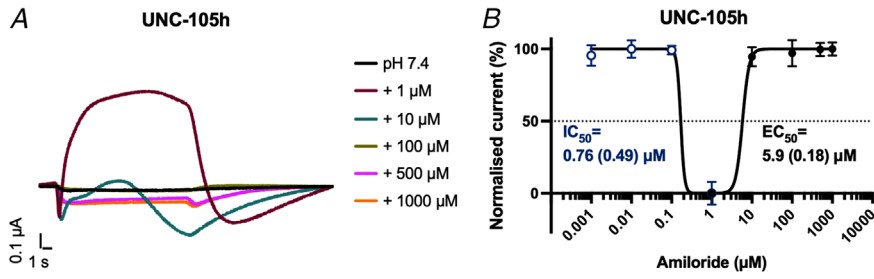

**Figure 9. Paradoxical effect of amiloride on UNC-105h**
transient currents at pH 7.4 can be blocked by amiloride. *A*, representative transient currents at increasing concentrations of amiloride. Amiloride blocks the channel at 1 µM concentration but other concentrations do not affect the currents. The *Xenopus* oocyte membrane was clamped at −60 mV. Raw currents in µA (*y*-axis), voltage steps in mV (*x*-axis) as indicated. *B*, amiloride dose response of UNC-105 with an $IC_{50}$ of 0.76 µM amiloride (SD = 0.49) and $EC_{50}$ of 5.9 µM amiloride (SD = 0.18). *N* = 7. Currents were recorded at a holding potential of −60 mV, baseline subtracted using pre-stimulus current and drift corrected, then normalised to the maximum current ($I_{max}$) calculated for each oocyte individually, and best fitted with the Hill equation (variable slope). Data points and error bars represent mean (SD). [Colour figure can be viewed at wileyonlinelibrary.com]

of dopaminergic neurons and enhances dopamine release required for associative learning (Voglis & Tavernarakis, 2008). Our electrophysiological characterisation of ASIC-1 in *Xenopus* oocytes supports the proposed working model of Voglis & Tavernarakis (2008) in which ASIC-1 at the presynaptic terminal is activated by a local drop in pH during the release of dopamine from the pre-synaptic terminal, which in turn promotes sustained dopaminergic signalling. Expression in head neurons or glia suggests a likely role for ACD-2 in modulating synaptic function, similar to that described for the *C. elegans* ASIC-1, or modulating neuronal function as described for the glial DELM-1, DELM-2 or ACD-1 (Han et al., 2013; Voglis & Tavernarakis, 2008; Wang et al., 2008). DEL-9 is expressed in the GABAergic motor neuron AVL, which synapses on to the enteric muscle and regulates the expulsion step of the defecation motor programme (McIntire et al., 1993), and the egg-laying muscles which are responsible for the expulsion of eggs.

This suggests that, in common with the acid-inhibited channels ACD-5 and FLR-1/ACD-3/DEL-5 (Kaulich et al., 2022), it could function in the coordination of rhythmic behaviours.

With the identification of four new members (ACD-5, DELM-1, DEL-4 and UNC-105h), we have expanded the acid-inhibited channels in *C. elegans*, in addition to the previously described ACD-1 (Wang et al., 2008). This shows that even on exposure to the same stimulus (here protons), these channels show a remarkable functional diversity, most likely responding to the demands in their local environment (i.e. in the intestinal lumen (Kaulich et al., 2022), duct of the kidney (Collier & Snyder, 2009) and the synaptic cleft (Du et al., 2014)). UNC-105 functions in the body wall muscle, and gain-of-function mutations that increase Na$^+$ influx cause hypercontraction, indicating a role in maintaining cell excitability (Park & Horvitz, 1986b; Garcia-Anoveros et al., 1998). Previous studies,

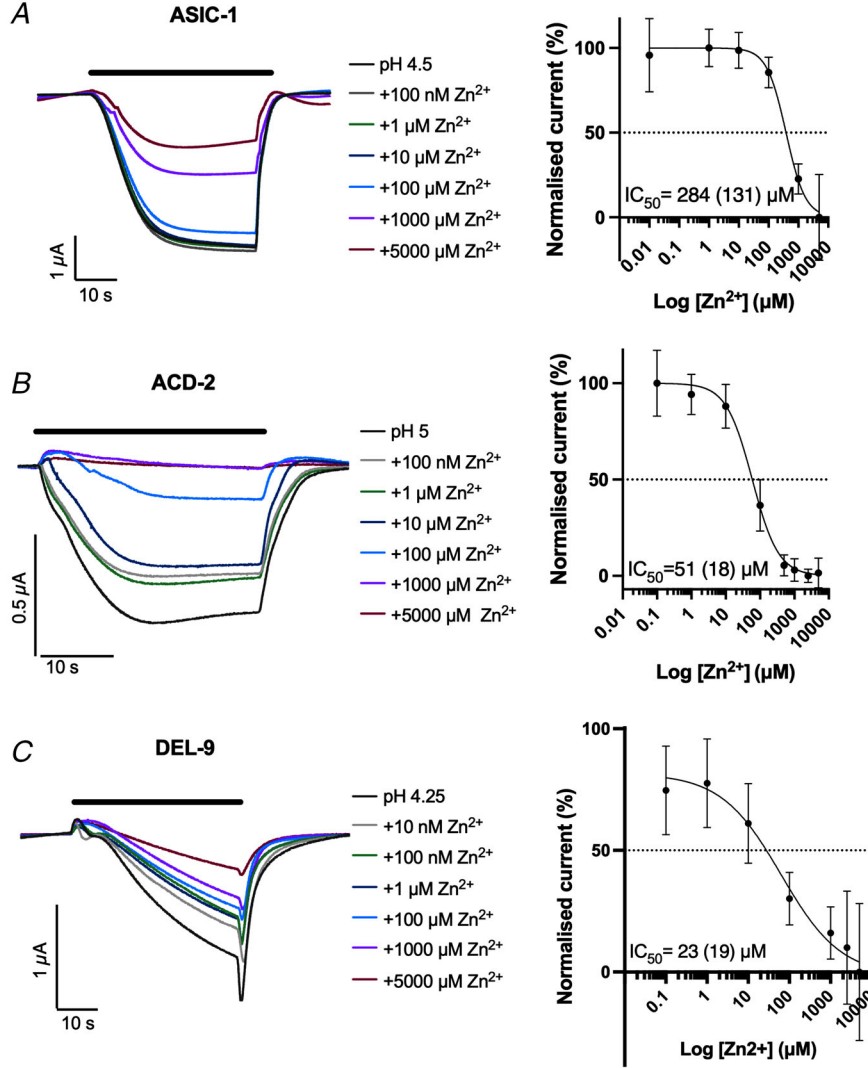

**Figure 10. Zinc modulation of acid-activated DEG/ENaC channels in *Xenopus* oocytes at pH$_{50}$**
Zn$^{2+}$ can block homomeric ASIC-1, ACD-2 and DEL-9 acid-evoked currents. Shown are representative example traces (left) and dose response curves (right) for each channel. ASCI-1 (*A*), ACD-2 (*B*) and DEL-9 (*C*) pH$_{50}$-evoked transient currents can be blocked by Zn$^{2+}$ in a dose-dependent manner (at pH$_{50}$ of each channel). Zn$^{2+}$ dose response of ASIC-1 with an IC$_{50}$ of 284 μM Zn$^{2+}$ (SD = 131, *N* = 10), ACD-2 with an IC$_{50}$ of 51 μM Zn$^{2+}$ (SD = 18, *N* = 15) and DEL-9 with an IC$_{50}$ of 23 μM Zn$^{2+}$ (SD = 19, *N* = 13). Baseline subtraction (using the pre-stimulus current, i.e. at pH 7.4) and drift correction was applied with Roboocyte2+ software. Currents were recorded at a holding potential of −60mV and are normalised to the maximum current (*I*$_{max}$) calculated for each oocyte individually, and best fitted with the Hill equation (variable slope). Data points and error bars represent mean (SD). The black bar indicates perfusion of the respective Zn$^{2+}$ concentration. [Colour figure can be viewed at wileyonlinelibrary.com]

involving electrophysiological recording from muscle, have already suggested that *C. elegans* body muscles are proton-sensitive (Jospin et al., 2004): under voltage-clamp and current-clamp conditions, decreasing external pH from 7.2 to 6.1 led to a reversible depolarization of muscle cells (Jospin & Allard, 2004). However, in an *unc-105* null mutant, the pH-sensitive current could still be observed, and acid-evoked depolarization is moreover suggestive of the involvement of an acid-activated,

muscle-expressed channel, such as DEL-9, rather than an acid-inhibited channel like UNC-105. The roles of acid-sensitive DEG/ENaCs in body muscle clearly merit further investigation.

Finally, DEG/ENaCs can also exert an effect on neuronal function from surrounding glia. Mutation of the *C. elegans* DEG/ENaC, *acd-1*, expressed in the amphid sheath cells, exacerbates these sensory deficits and deficits caused by mutations of genes implicated in sensory functions

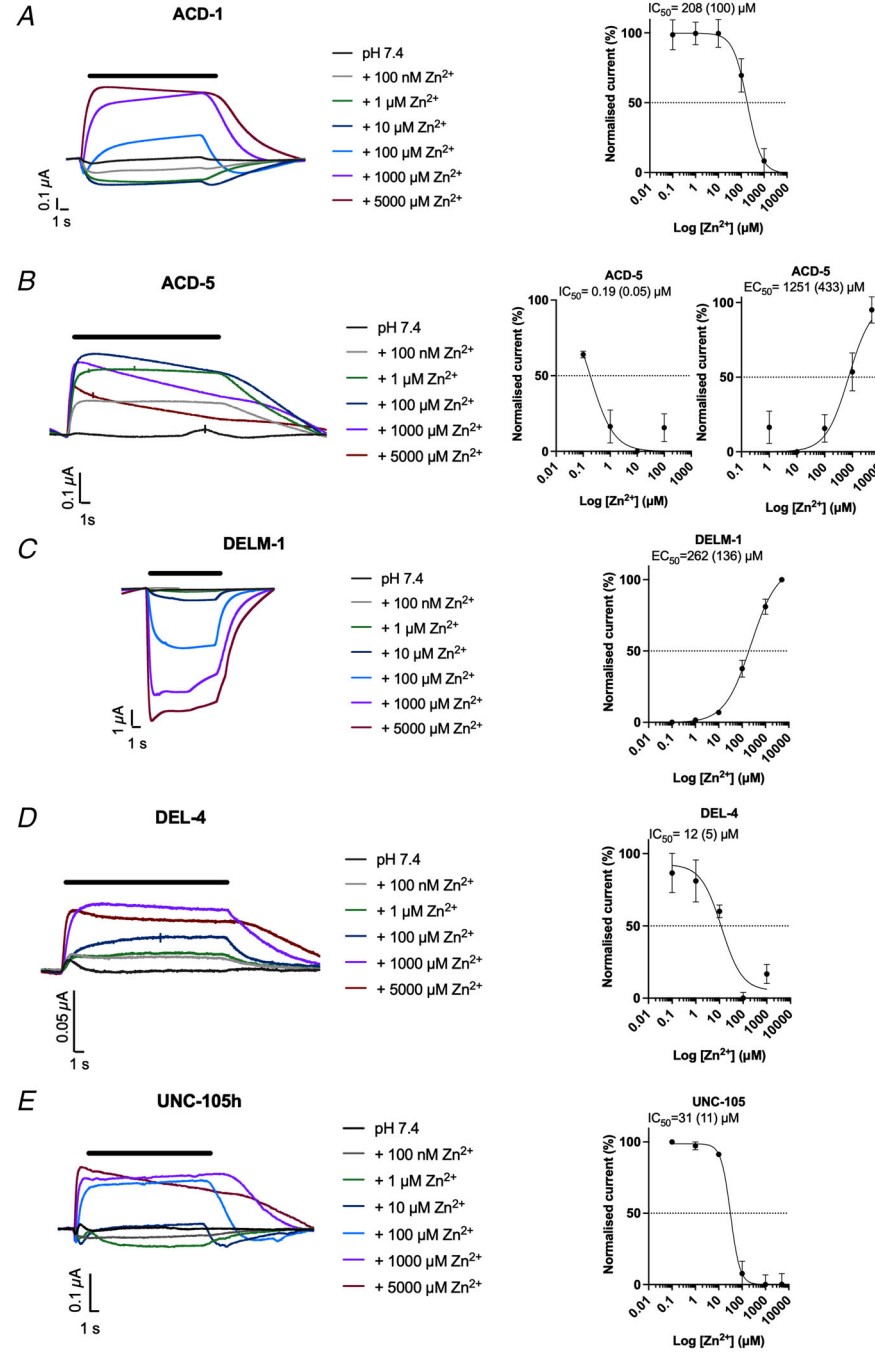

**Figure 11. Zinc modulation of acid-inhibited DEG/ENaC channels in *Xenopus* oocytes at pH 7.4**

*A–E*, $Zn^{2+}$ modulates homomeric ACD-1, ACD-5, DELM-1, DEL-4, and UNC-105 currents at pH 7.4 in a dose-dependent manner. Shown are representative example traces (Left) and dose–response curves (right) for each channel: $Zn^{2+}$ dose response of ACD-1 with an $IC_{50}$ of 208 μM (SD = 100, $N$ = 7) $Zn^{2+}$, ACD-5 with an $IC_{50}$ of 0.19 μM (SD = 0.05, $N$ = 4) and an $EC_{50}$ of 1251 μM (SD = 433, $N$ = 4), DELM-1 with an $EC_{50}$ of 262 μM (SD = 136, $N$ = 4), DEL-4 with an $IC_{50}$ of 12 μM (SD = 5, $N$ = 5), and UNC-105h with an $IC_{50}$ of 31 μM (SD = 11, $N$ = 4) $Zn^{2+}$. Baseline subtraction and drift correction was applied with Roobocye2+ software. Currents were recorded at a holding potential of −60 mV and are normalised to the maximum current ($I_{max}$) calculated for each oocyte individually, and best fitted with the Hill equation (variable slope). Data points and error bars represent mean (SD). The black bar indicates perfusion of the respective $Zn^{2+}$ concentration. [Colour figure can be viewed at wileyonlinelibrary.com]

in other amphid neurons (Wang et al., 2008, 2012). Artificially increasing intracellular Ca$^{2+}$ levels in one of these neurons bypassed the need for ACD-1, supporting the idea that ACD-1 modulates neuronal excitability (Wang et al., 2008, 2012). This idea is supported by the second example: DELM-1 (which we have shown is acid-inhibited, like ACD-1) and DELM-2 are expressed in the glia cells associated with nose touch neurons, an the OLQ and IL1 neurons, on which they appear to exert a similar effect (Han et al., 2013). Vertebrate ASICs are also expressed in glia and, for example, some of the roles identified in learning may in fact be glia-based (Hill & Ben-Shahar, 2018), so disentangling glial from neuronal functions represents an exciting avenue for future investigations. In summary, *C. elegans* acid-sensitive ion channels appear to be involved in regulating the general excitability of a wide range of cell types, including muscle, glia, epithelia and neurons, and in a wide range of functional contexts. We have also shown that many of the channels show a dual or bidirectional response to stimuli, depending on the concentration. This might reflect environmental contexts (i.e. the channel does not encounter this kind of concentration under physiological conditions) but it could also reflect a mechanism to fine-tune responses to a combination of compounds which might fit well with the proposed model of ASICs acting in 'coincidence detection' (Bohlen et al., 2011). Precise channel localisation (for example, to identify localisation at specific synapses) and correlation with behavioural testing are the necessary next steps for exploring *in vivo* function.

The expanded group of *C. elegans* DEG/ENaCs thus encompasses a huge variety of channel properties, with respect to proton-dependence profiles, their interactions with amiloride, zinc and NSAIDs, and mechanosensitivity (Chalfie & Sulston, 1981; Chatzigeorgiou et al., 2010; Fechner et al., 2021; Geffeney et al., 2011). These distinct functional capabilities do not necessarily cluster with overall sequence similarity. For example, in our phylogram (Fig. 1), whereas ACD-1, ACD-5 and DELM-1 cluster closely to each other, the two other acid-inhibited subunits, DEL-4 and UNC-105, are closer to ASIC-1 and the mechanosensitive members, and none of the three acid-activated subunits cluster together. Disentangling the molecular basis of this diversity of function, and comparison across phyla, represents an important avenue for better understanding structure–function relationships of DEG/ENaCs. For instance, solving the full-length structure of the chicken ASIC1a, including the N-terminal extension, and structure guided sequence

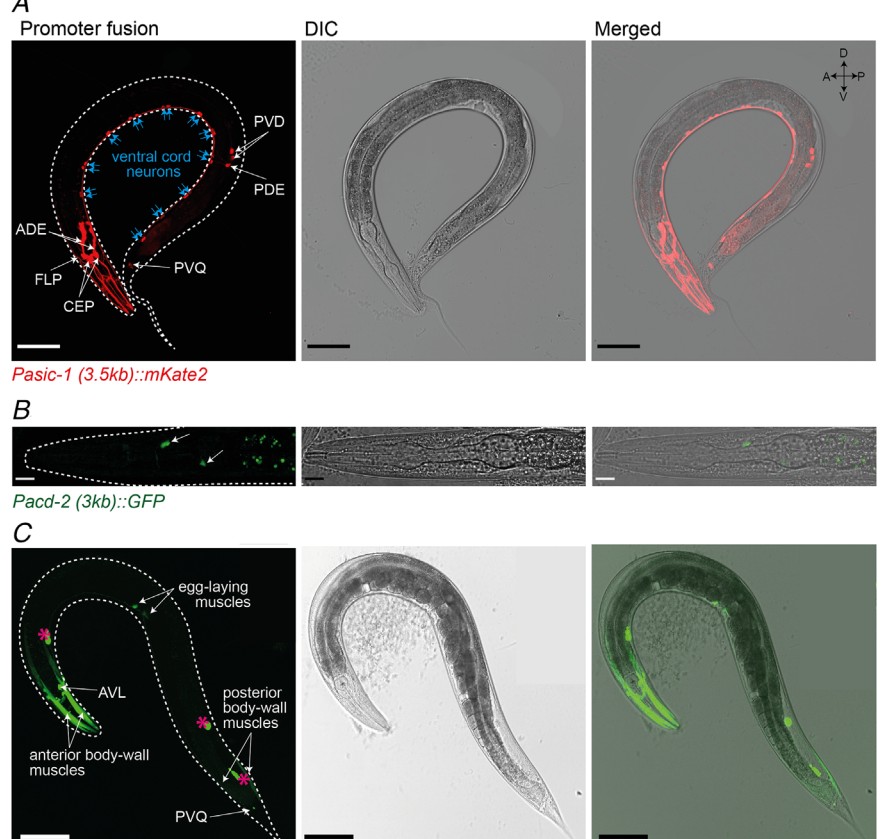

**Figure 12. Acid-activated DEG/ENaCs are expressed in neuronal and non-neuronal tissue**

Expression pattern of the transcriptional promoter fusions of the acid-activated DEG/ENaC genes with *mKate2* or *GFP* in L4 and young adults. *A*, the *asic-1* promoter drives expression in the dopaminergic neurons and PVDs and FLPs. *B*, *acd-2* promoter expression can be faintly detected in the head, which based on localisation and RNA-sequencing data by Cao et al. (2017) could be neurons or glia cells. Green dots in the intestine are autofluorescence. *C*, The *del-9* promoter drives expression in the body-wall and egg-laying muscles, in head neurons, PVQ neuron and the GABAergic neuron AVL. Pink asterisks indicate the coelomocytes (*Punc-122::GFP*, used as co-injection marker to select transgenic animals). Scale bars: 100 μm (*A*), 10 μm (*B*) and 100 μm (*C*). Orientation indicator shown top right, A, anterior; D, dorsal; P, posterior; V, ventral. [Colour figure can be viewed at wileyonlinelibrary.com]

**Table 4. Summary of physiological properties of *C. elegans* acid-sensitive DEG/ENaCs**

| DEG/ENaC | $pH_{50}$ | Selectivity sequence | Amiloride | $Zn^{2+}$ | Expression | Reference |
|---|---|---|---|---|---|---|
| **Acid inhibited** | | | | | | |
| ACD-1 | 6.4 | $Na^+ > Li^+ > K^+$ | $IC_{50} = 99\ \mu M$ | $IC_{50} = 208\ \mu M$ | Glia | Wang et al. (2008) |
| ACD-5 | 4.87 (in) 6.48 (act) | $Li^+ > K^+ = Na^+$ | $IC_{50} = 131\ \mu M$ | $IC_{50} = 0.19\ \mu M$ $EC_{50} = 1251\ \mu M$ | Intestine | Kaulich et al. (2022) |
| DEL-4 | 5.7 | $Na^+ = Li^+ > K^+$ | $IC_{50} = 179\ \mu M$ | $IC_{50} = 12\ \mu M$ | Neurons | D. Petratou, M. Gjikolaj, E. Kaulich, W. R. Schafer, N. Tavernarakis, unpublished observations). |
| DELM-1 | 5.50 | $Li^+ > Na^+ > K^+$ | $IC_{50} = 120\ \mu M$ | $EC_{50} = 262\ \mu M$ | Glia | Han et al. (2013) |
| UNC-105 | 6.30 | $Li^+ > Na^+ = K^+$ | $IC_{50} = 0.76 \mu M$ $EC_{50} = 5.9 \mu M$ | $IC_{50} = 31\ \mu M$ | Muscle | Garcia-Anoveros et al. (1998), Jospin & Allard (2004) |
| **Acid activated** | | | | | | |
| ACD-2 | 5.04 | $Na^+ > Li^+ > K^+$ | $IC_{50} = 87\ \mu M$ | $IC_{50} = 51\ \mu M$ | Head neurons/ glia | |
| ASIC-1 | 4.50 | $K^+ = Li^+ > Na^+$ | $IC_{50} = 108\ \mu M$ | $IC_{50} = 284\ \mu M$ | Neurons | |
| DEL-9 | 4.33 | $Na^+ = Li^+ = K^+$ | Acid-induced currents are insensitive | $IC_{50} = 23\ \mu M$ | Muscle, neurons | |

Table showing the channel properties described here, incorporated with those reported in the references indicated. act, activation; in, inhibition.

comparison with other members revealed conserved residues important for ion selectivity and gating (Yoder & Gouaux, 2020). We have provided evidence here that the 'GAS' belt is important for functioning of the channels, in line with previous research (Baconguis et al., 2014; Carattino & Della Vecchia, 2012; Chen et al., 2022; Kellenberger, Gautschi et al., 1999; Li, Yang et al., 2011; Lynagh et al., 2017, 2020).

Many DEG/ENaC/ASICs can form heteromeric channels, and therefore a future avenue to explore is how co-expression of subunits influences channel properties. We have shown non-selective acid-sensitive currents for the DEL-9 heteromer; while this could be a primordial feature of DEG/ENaCs, as previously suggested (Dürrnagel et al., 2012), it might also be that other subunits are required for ion permeability. Previous research has shown that ion-selectivity and permeability is influenced by subunit composition. For instance, ASIC and ENaC subunits can change ion permeability of the channel (Vallee et al., 2021). The most recent structure of the chicken ASIC1a has suggested that His29 stabilises the pre-TM1 re-entrant loop and lower pore via hydrogen bonding interactions with the 'GAS' belt residues on neighbouring subunits (Yoder & Gouaux, 2020) and both motifs from each subunit are implicated in gating and ion selectivity (Chen et al., 2022; Grunder et al., 1999; Kellenberger, Gautschi et al., 1999; Kellenberger,

Hoffmann-Pochon et al., 1999; Kellenberger & Schild, 2002; Kellenberger et al., 2001, 2003).

Likewise, correlating protein sequence with functional differences or similarities will help in elucidation of the molecular basis of function. For instance, the *C. elegans* acid-sensitive DEG/ENaCs do not desensitise, nor does the lamprey ASIC1 (Li et al., 2010; Li, Yang et al., 2011) despite sharing 68% identity and almost 80% similarity with the chicken ASIC1a. The kinetics of the lamprey ASIC1 has been narrowed down to one amino acid, Trp64, which corresponds to Arg65 in the chicken ASIC1a (Li et al., 2010). Interestingly, for all acid-sensing DEG/ENaCs described here the corresponding residue is hydrophobic (ACD-2 Leu124, ASIC-1 Ile58, DEL-9 Ile125, ACD-1 Leu124, ACD-5 Leu155, DEL-4 Leu75, DELM-1 Leu115, UNC-105 Leu75). Changes from a polar to a hydrophobic residue may alter the structure and prevent desensitisation. Based on the structure of the chicken ASIC1a, Arg65 in helix1 comes in close contact with the Glu426 situated at the border of the transmembrane domain (TMD) and extracellular domain (ECD) in the 'wrist' subdomain. This location of basic and acid residues leads to the formation of a salt bridge, which couples the TMD and ECD. Having a hydrophobic residue (Leu, Ile or Trp) in place of Arg, as in the *C. elegans* or lamprey subunits, results in the loss of the salt bridge and loss of coupling of TMD and ECD, which might explain the non-desensitising currents of lamprey ASIC1

and the *C. elegans* acid-sensing DEG/ENaCs during acid-stimulation.

Finally, our comprehensive electrophysiological characterisation also provides a foundation for compound screens against ASICs for drug discovery, both in the worm and in *Xenopus* oocytes. Anti-helminthic drugs described to date act on ion channels in neurons and muscles. Our characterisation of *C. elegans* acid-sensitive DEG/ENaCs revealed expression in these tissues, opening up new avenues for future investigation of DEG/ENaCs as potential anti-helminthic targets in parasitic relatives of *C. elegans*.

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

## Additional information

### Data availability statement

All data generated or analysed during this study are included in the manuscript.

### Competing interests

The authors have no competing interests to declare.

### Author contributions

E.K., D.S.W. and W.R.S. conceived the experiments. E.K. performed and analysed all experiments, except cloning of, and pilot TEVC experiments for, *egas-1, egas-2, egas-3, and egas-4* (performed by PTNM). E.K. analysed the data. E.K. and D.S.W. wrote the manuscripts, E.K., D.S.W., P.T.N.M., and W.R.S. edited the manuscript. WRS acquired funding. All authors approved the final version of the manuscript and agree to be accountable for all aspects of the work in ensuring that questions related to the accuracy or integrity of any part of the work are appropriately investigated and resolved. All persons designated as authors qualify for authorship, and all those who qualify for authorship are listed.

### Funding

This work was supported by the Medical Research Council, as part of United Kingdom Research and Innovation (also known as UK Research and Innovation) [MRC file reference number MC-A023-5PB91], by the Wellcome Trust [grant reference number WT103784MA] and by the National Institutes of Health [grant reference numbers R01NS110391 and R21DC015652], all to WRS. For the purpose of Open Access, the MRC Laboratory of Molecular Biology has applied a CC BY public copyright licence to any Author Accepted Manuscript (AAM) version arising from this submission. The funders had no role in study design, data collection and interpretation, or the decision to submit the work for publication.

### Acknowledgements

The authors are very grateful to Kyuhyung Kim's lab (DGIST) for providing us with their *C. elegans* DEG/ENaC transcriptional reporter plasmids and sharing their unpublished expression data with us. The authors thank members of the Schafer, Taylor and de Bono labs (MRC LMB), Beets and Temmerman (KU Leuven) and Pless (University of Copenhagen) labs, Iris Hardege, Vikram B. Kasaragod (MRC LMB), and Ewan St. John Smith (University of Cambridge) for helpful discussions. The authors are grateful to the LMB support facilities, in particular Ben Sutcliffe, Jonathan Howe, and Nick Barry from the Light Microscopy Facility, and Sue Hubbard, Mark Cussens, and Martyn Howard and their team for preparing solutions and NGM plates.

### Keywords

acid-sensing ion channel, amiloride, degenerin/epithelial sodium channel, proton-gated, zinc

### Supporting information

Additional supporting information can be found online in the Supporting Information section at the end of the HTML view of the article. Supporting information files available:

**Statistical Summary Document**
**Peer Review History**

