## [Peer Review History · The Journal of Physiology]

Proton-gated; acid-sensing ion channels Physiological insight into the conserved properties of *Caenorhabditis elegans* acid-sensing DEG/ENaCs

Eva Kaulich, Patrick TN McCubbin, William R Schafer, and Denise S Walker

DOI: 10.1113/JP283238

Corresponding author(s): Denise Walker (dwalker@mrc-lmb.cam.ac.uk)

Review Timeline:

Submission Date:	30-Apr-2022
Editorial Decision:	09-Jun-2022
Revision Received:	28-Jul-2022
Editorial Decision:	18-Aug-2022
Revision Received:	16-Sep-2022
Accepted:	28-Sep-2022

Senior Editor: *Peying Fong*

Reviewing Editor: *Morag Mansley*

Transaction Report:

Dear Dr Walker,

Re: JP-RP-2022-283238 "Proton-gated; acid-sensing ion channels. Physiological insight into the conserved properties of *Caenorhabditis elegans* acid-sensing DEG/ENaCs" by Eva Kaulich, Patrick TN McCubbin, William R Schafer, and Denise S Walker

Thank you for submitting your manuscript to The Journal of Physiology. It has been assessed by a Reviewing Editor and by 2 expert Referees and I am pleased to tell you that it is considered to be acceptable for publication following satisfactory revision.

The reports are copied at the end of this email. Please address all of the points and incorporate all requested revisions, or explain in your Response to Referees why a change has not been made.

NEW POLICY: In order to improve the transparency of its peer review process The Journal of Physiology publishes online as supporting information the peer review history of all articles accepted for publication. Readers will have access to decision letters, including all Editors' comments and referee reports, for each version of the manuscript and any author responses to peer review comments. Referees can decide whether or not they wish to be named on the peer review history document.

Authors are asked to use The Journal's premium BioRender (<https://biorender.com/>) account to create/redraw their Abstract Figures. Information on how to access The Journal's premium BioRender account is here: <https://physoc.onlinelibrary.wiley.com/journal/14697793/biorender-access> and authors are expected to use this service. This will enable Authors to download high-resolution versions of their figures. The link provided should only be used for the purposes of this submission. Authors will be charged for figures created on this premium BioRender account if they are not related to this manuscript submission.

I hope you will find the comments helpful and have no difficulty returning your revisions within 4 weeks.

Your revised manuscript should be submitted online using the links in Author Tasks: Link Not Available.

Any image files uploaded with the previous version are retained on the system. Please ensure you replace or remove all files that have been revised.

REVISION CHECKLIST:

- Article file, including any tables and figure legends, must be in an editable format (eg Word)
- Abstract figure file (see above)
- Statistical Summary Document
- Upload each figure as a separate high quality file
- Upload a full Response to Referees, including a response to any Senior and Reviewing Editor Comments;
- Upload a copy of the manuscript with the changes highlighted.

- A potential 'Cover Art' file for consideration as the Issue's cover image;
- Appropriate Supporting Information (Video, audio or data set https://jp.msubmit.net/cgi-bin/main.plex?form_type=display_requirements#supp).

To create your 'Response to Referees' copy all the reports, including any comments from the Senior and Reviewing Editors, into a Word, or similar, file and respond to each point in colour or CAPITALS and upload this when you submit your revision.

I look forward to receiving your revised submission.

If you have any queries please reply to this email and staff will be happy to assist.

Yours sincerely,

Dr Peiyong Fong
Senior Editor
The Journal of Physiology
<https://jp.msubmit.net>
<http://jp.physoc.org>
The Physiological Society
Hodgkin Huxley House
30 Farringdon Lane
London, EC1R 3AW
UK
<http://www.physoc.org>
<http://journals.physoc.org>

REQUIRED ITEMS:

- Author photo and profile. First (or joint first) authors are asked to provide a short biography (no more than 100 words for one author or 150 words in total for joint first authors) and a portrait photograph. These should be uploaded and clearly labelled with the revised version of the manuscript. See Information for Authors for further details.
- Your manuscript must include a complete Additional Information section.
- The Journal of Physiology funds authors of provisionally accepted papers to use the premium BioRender site to create high resolution schematic figures. Follow this link and enter your details and the manuscript number to create and download figures. Upload these as the figure files for your revised submission. If you choose not to take up this offer we require figures to be of similar quality and resolution. If you are opting out of this service to authors, state this in the Comments section on the Detailed Information page of the submission form. The link provided should only be used for the purposes of this submission. Authors will be charged for figures created on this premium BioRender account if they are not related to this manuscript submission.
- Please upload separate high-quality figure files via the submission form.
- Please ensure that any tables are in Word format and are, wherever possible, embedded in the article file itself.
- Please ensure that the Article File you upload is a Word file.
- Your paper contains Supporting Information of a type that we no longer publish. Any information essential to an understanding of the paper must be included as part of the main manuscript and figures. The only Supporting Information that we publish are video and audio, 3D structures, program codes and large data files. Your revised paper will be returned to you if it does not adhere to our Supporting Information Guidelines
- A Statistical Summary Document, summarising the statistics presented in the manuscript, is required upon revision. It must be on the Journal's template, which can be downloaded from the link in the Statistical Summary Document section here: https://jp.msubmit.net/cgi-bin/main.plex?form_type=display_requirements#statistics
- Papers must comply with the Statistics Policy https://jp.msubmit.net/cgi-bin/main.plex?form_type=display_requirements#statistics

In summary:

- If $n \leq 30$, all data points must be plotted in the figure in a way that reveals their range and distribution. A bar graph with data points overlaid, a box and whisker plot or a violin plot (preferably with data points included) are acceptable formats.
- If $n > 30$, then the entire raw dataset must be made available either as supporting information, or hosted on a not-for-profit repository e.g. FigShare, with access details provided in the manuscript.
- 'n' clearly defined (e.g. x cells from y slices in z animals) in the Methods. Authors should be mindful of pseudoreplication.
- All relevant 'n' values must be clearly stated in the main text, figures and tables, and the Statistical Summary Document (required upon revision)
- The most appropriate summary statistic (e.g. mean or median and standard deviation) must be used. Standard Error of the Mean (SEM) alone is not permitted.

- Exact p values must be stated. Authors must not use 'greater than' or 'less than'. Exact p values must be stated to three significant figures even when 'no statistical significance' is claimed.

- Statistics Summary Document completed appropriately upon revision

- Please include an Abstract Figure. The Abstract Figure is a piece of artwork designed to give readers an immediate understanding of the research and should summarise the main conclusions. If possible, the image should be easily 'readable' from left to right or top to bottom. It should show the physiological relevance of the manuscript so readers can assess the importance and content of its findings. Abstract Figures should not merely recapitulate other figures in the manuscript. Please try to keep the diagram as simple as possible and without superfluous information that may distract from the main conclusion(s). Abstract Figures must be provided by authors no later than the revised manuscript stage and should be uploaded as a separate file during online submission labelled as File Type 'Abstract Figure'. Please ensure that you include the figure legend in the main article file. All Abstract Figures should be created using BioRender. Authors should use The Journal's premium BioRender account to export high-resolution images. Details on how to use and access the premium account are included as part of this email.

EDITOR COMMENTS

Reviewing Editor:

The review of the manuscript, "Proton-gated; acid sensing ion channels. Physiological insight into the conserved properties of *Caenorhabditis elegans* acid-sensing DEG/ENaCs" that was submitted to The Journal of Physiology is complete, having been assessed by 2 referees as well as the reviewing and senior editors.

The Editors have carefully read your manuscript and considered the points raised by the referees. At present the manuscript requires revisions throughout, with some possible experiments carried out to satisfy key points raised. In particular, both referees have noted that appropriate controls to account for endogenous currents are lacking. The major points listed by the referees will need to be addressed as currently the paper is deemed to have the potential to be of influence but not in its current form.

We therefore recommend to Provisionally Accept, but with clear instruction that the referees' concerns/suggestions are addressed.

No p values have been reported.

No statistical analyses have been carried out, as per the first major point from referee 1.

Senior Editor:

Many thanks for submitting your study in response to the Call for Papers. Attached you will see the reports from two Expert Referees and a summary from the Reviewing Editor. I concur with the Reviewing Editor's perspective.

As you see, both Referees voice concerns about interpretation of data as well as adequate powering of controls. The absence of adequate controls renders the interpretations (and hence, conclusions) less convincing to readers. These data may in fact already be available, but we trust their acquisition is possible before the time typically granted for revised manuscripts (4 weeks). Addressing this matter, as well as providing rigorous statistical tests, will be particularly critical.

Overall, in your revised manuscript I strongly encourage you to apply greater rigor in analysis and actively place your findings within the broader scope of the existing literature with a critical eye.

REFEREE COMMENTS

Referee #1:

In this manuscript Kaulich and colleagues analyze the electrophysiological properties of *C. elegans* DEG/ENaC channels expressed in the *Xenopus* oocytes system. In particular, the authors focus on the pH dependence of these channels as well as the sensitivity to the anti-hypertensive drug amiloride and to the ion zinc. The study is relevant for the field because *C. elegans* is a genetically amenable system which allows for straightforward *in vivo* studies that link protein function to organismal behavior. In particular, the pH dependence of DEG/ENaC channels has been linked to disease states in humans but a direct correlation between protein function and manifestation of the disease has been more difficult to tease out in complex systems like the mouse. The study could be useful, but the manuscript has several main issues.

1. The main problem with this manuscript is that the authors fail to recognize that all the channels shown in figure 3 panel E are most likely non-functional or not processed to the oocytes cell membrane or not activated in these recordings. There is

no indication whatsoever in their data that the currents shown are something more than endogenous oocytes currents. As a consequence, this entire panel and related panels in the following figures, should be removed or moved to supplementary data, or these negative results should be simply mentioned in the text.

2. Mec-4 and mec-10 data are somewhat concerning. Goodman et al. 2002 and Bianchi et al. 2004 showed that wild type mec-4 produces very small currents when expressed in *Xenopus* oocytes (<0.05 uA). The authors are showing a few data points that are above 0.5 uA, suggesting that in this case they are reporting oocytes endogenous currents. Wild type mec-10 does not produce currents when expressed in oocytes, as it functions as an accessory subunit. Yet the authors are reporting several data points with substantial currents. Again, there is concern here that the authors are reporting endogenous oocytes currents.

3. Why is there one single oocyte that expresses deg-1 that has substantial current? Is this again endogenous current?

4. Del-9 currents are insensitive to amiloride and non-selective. I have no confidence that these currents are actually due to expression of del-9. There is an endogenous current in *Xenopus* oocytes that has exactly the same kinetics as the currents shown in figure 6 D and which may be pH and zinc sensitive. They also need to keep in mind that this endogenous current is variable from batch to batch of oocytes and that it is more prominent in oocytes that fail to express the protein of interest. The authors need to perform additional experiments in support of their conclusion that the currents reported are indeed del-9. Point mutations that change the channel selectivity or that render the channel nonfunctional might be useful here.

5. I have similar concerns on unc-105 currents. Especially the currents shown in figure 7 are not convincing and could be endogenous. Have the authors tried to inject higher concentrations of RNA or record currents earlier or later after the injection? Larger currents, clearly above the level of endogenous current for that batch of oocytes would be more convincing. Analysis of the selectivity and/or point mutation that disrupt selectivity or function would also be needed here.

6. Figure 4: the dose response curves have quite large SD and lack a point or two around pH 6.5 that would give more confidence that the estimate of the p_{H0.5} is correct.

7. In figure 4 panel A, delm-1 currents seem to be stimulated by pHs 7.5 and 6 as compared to 8. The authors should discuss this.

8. Figure 12 panel A: zinc seems to have a dual effect on ACD-1 depending on the concentration. The authors need to discuss this.

9. Since delm-1 and delm-2 are expressed in the same cells, it would be interesting to determine whether delm-2 regulates delm-1 pH sensitivity.

Minor:

1. Figure 2, the yellow circles are hard to see.

2. Line 314: the term "excitatory" is normally used in reference to a synapse or a current. It might be better to use a different term here.

3. Line 357: "Actual current..." this sentence is not clear. What do the authors mean by "actual current"?

4. Line 391: "other DEG/ENaC subunits.." which ones? The authors need to list them.

5. Line 403: "high concentrations of Zn²⁺". How high? The authors need to provide a range. High is a vague term.

6. Line 621: acd-1 is expressed in the Amphid sheath cells, not socket cells.

Referee #2:

This manuscript by Kaulich et al. aimed to investigate and characterize potential DEG/ENaC ion channels in *C. elegans*. Given the importance of DEG/ENaC channels in many vertebrate physiological processes and the experimental accessibility of *C. elegans* as a model organism, this study is of interest to a broad readership. However, this reviewer has several concerns which should be addressed.

Major concerns:

1. The entire manuscript lacks statistical analyses of the presented data.

2. Some parts of the manuscript read like a review article. For example, Results, Lines 301-307. The manuscript should be rewritten to meet the formal research article guidelines.

3. Figures 1 and 2 should be placed and described in the results section. Please indicate in the legend of Figure 1 that

protein sequences were used to construct the tree. Were the database entries verified? Did the protein sequences match the genomic information available in the databases (there can be discrepancies between coding sequences and amino acid sequences). Please indicate the used acronyms as well (e.g., lower case letters indicating species of one group).

4. I struggle understanding the data presented in Figure 3. Positive current values are shown, indicative of outward currents. This is quite surprising given that vertebrate DEG/ENaCs are cation channels that usually display inward currents. There is also a discrepancy with the data presented in Figure 5, where e.g., ACD-2 shows clear inward currents at -60 mV. Please provide original current traces for the investigated ion channel candidates.

5. Many of the investigated DEG/ENaC channels do not show pH-sensitive currents (Figure 3) and transmembrane currents are within the magnitude of water-injected controls. From these data one cannot conclude that these proteins do not form functional (Acid-sensitive) ion channels. Appropriate controls showing that the proteins are indeed present in the oocyte membranes are needed to support the electrophysiology data.

6. Methods, Line 193: There appears to be an error regarding the injection volume. I doubt 25 microl would fit into one oocyte.

7. For the presented electrophysiology data, please indicate 0 current levels in order to allow justification of inward/outward currents.

8. Results, Line 321-331: I do not fully understand the Erev shift experiments. If extracellular ions are substituted and the ion channel is conducting the substitutes, it is not surprising to obtain a shift in Erev. But does it indicate a "preference" or "selectivity" as stated in the manuscript? It would also be helpful to provide the estimated Nernst potentials for the experimental conditions.

9. Figure 8: Please explain how the dose-response curve was plotted, given the activation/inhibition that is dependent on the amiloride concentration.

10. Many representative traces/figure panels (e.g. of the I/V experiments) are very small and very difficult to read.

Minor concerns:

1. Introduction Lines 50-66: Some amphibian ENaCs have a much stronger pH-sensitivity than human ENaCs and there is somewhat conflicting data on the pH-values that activate human ENaCs (e.g. PMID: 31248986). For ASICs, they are not only open or closed, they also show desensitization and a steady state desensitization (e.g. PMID: 35201495). This should be included in the introduction as well.

2. Table 1: Do all accession numbers refer to Uniprot? The methods suggest that NCBI and wormbase are also used.

3. Methods, Lines 197/198: What was the rationale for filling the electrodes with a mixture of KCl and K-acetate?

4. Methods, Line 205: Please clarify what is meant by "within the error of 210 mOsm". Does this mean that 210 mOsm changes were accepted?

5. The reference "Petratou et al., submitted" should be removed - this is not a citable source.

6. Table 3: "in" or "ex" are mentioned in the legend but these are not used in the table.

7. There are quite a few statements relating to earlier work by the authors which are not supported by references.

END OF COMMENTS

Confidential Review

30-Apr-2022

28th July 2022

Dear reviewers and editors,

Thank you very much for your insightful and constructive comments. We hope that you will agree that our manuscript is much improved, thanks to your input. We have addressed these comments individually below, outlining in purple the changes that we have made.

Best wishes,

Denise Walker.

EDITOR COMMENTS

Reviewing Editor:

The review of the manuscript, "Proton-gated; acid sensing ion channels. Physiological insight into the conserved properties of *Caenorhabditis elegans* acid-sensing DEG/ENaCs" that was submitted to *The Journal of Physiology* is complete, having been assessed by 2 referees as well as the reviewing and senior editors.

The Editors have carefully read your manuscript and considered the points raised by the referees. At present the manuscript requires revisions throughout, with some possible experiments carried out to satisfy key points raised. In particular, both referees have noted that appropriate controls to account for endogenous currents are lacking. The major points listed by the referees will need to be addressed as currently the paper is deemed to have the potential to be of influence but not in its current form.

We therefore recommend to Provisionally Accept, but with clear instruction that the referees' concerns/suggestions are addressed.

No p values have been reported.

No statistical analyses have been carried out, as per the first major point from referee 1.

Senior Editor:

Many thanks for submitting your study in response to the Call for Papers. Attached you will see the reports from two Expert Referees and a summary from the Reviewing Editor. I concur with the Reviewing Editor's perspective.

As you see, both Referees voice concerns about interpretation of data as well as adequate powering of controls. The absence of adequate controls renders the interpretations (and hence, conclusions) less convincing to readers. These data may in fact already be available, but we trust their acquisition is possible before the time typically granted for revised manuscripts (4 weeks). Addressing this matter, as well as providing rigorous statistical tests, will be particularly critical.

Overall, in your revised manuscript I strongly encourage you to apply greater rigor in analysis and actively place your findings within the broader scope of the existing literature with a critical eye.

Please see our replies to the reviewers' comments below for details of how we have addressed them. Many of the reviewer comments related to Figure 3, where the responses of all the expressed channels were presented. In some cases, the reviewers expressed concern that we were overinterpreting negative results and requested that we leave out the data or put it in a supplemental figure. Since the latter option is not allowed by the journal, our preference has been to retain the figure but clarify our interpretation of the data. Namely, in cases where we did not observe currents, we do not know whether the channel gene was not well-expressed in the oocytes, or whether it does not form channels as a monomer, or whether it does form channels that are not active under the conditions tested. In cases (e.g. DEG-1) where the currents were small or of low reproducibility, we did not draw further conclusions but thought it would be of use nonetheless to show the full data. In general, we have felt it better practice to show all our results rather than selectively edit it on the basis of previous expectations. We are happy to discuss this issue further if questions arise.

We have added statistical analyses, where relevant, for Figures 3, 4 and 6, along with p values.

We absolutely agree that we could have better placed our findings in the context of observations for other members of the DEG/ENaC family. We have added several sections in the Results to do this better, for example, in relation to DELM-1/DELM-2 co-expression (line 368); UNC-105 permeability (line 386); DEL-9's lack of selectivity (line 411); lack of desensitisation, as also seen for lamprey ASIC1 (line 676).

REFeree COMMENTS

Referee #1:

In this manuscript Kaulich and colleagues analyze the electrophysiological properties of *C. elegans* DEG/ENaC channels expressed in the *Xenopus* oocytes system. In particular, the

authors focus on the pH dependence of these channels as well as the sensitivity to the anti-hypertensive drug amiloride and to the ion zinc. The study is relevant for the field because *C. elegans* is a genetically amenable system which allows for straightforward in vivo studies that link protein function to organismal behavior. In particular, the pH dependence of DEG/ENaC channels has been linked to disease states in humans but a direct correlation between protein function and manifestation of the disease has been more difficult to tease out in complex systems like the mouse. The study could be useful, but the manuscript has several main issues.

1. The main problem with this manuscript is that the authors fail to recognize that all the channels shown in figure 3 panel E are most likely non-functional or not processed to the oocytes cell membrane or not activated in these recordings. There is no indication whatsoever in their data that the currents shown are something more than endogenous oocytes currents. As a consequence, this entire panel and related panels in the following figures, should be removed or moved to supplementary data, or these negative results should be simply mentioned in the text.

Yes, we absolutely agree, the absence of current does not mean that the channel is not pH-sensitive, nor can any conclusion be drawn about the localisation on the membrane of the oocyte. We are not trying to claim anything other than that we tried to express the channels and did not observe currents, but we should have made this clearer in the text. Unfortunately, the Journal does not allow supplementary figures (but we agree that this would be a good supplementary figure).

Figure 3 shows our initial comprehensive screen of the *C. elegans* DEG/ENaC family members. We feel that the small currents/lack of current when expressing these monomers is useful information for others, as well as serving as additional internal controls (demonstrating, for example, that in a very large number of experiments, expressing diverse channels, we do not observe any evidence of endogenous currents resembling those seen for DEL-9). We therefore feel that it is important to retain them in figure 3 but, to take account of the referee's comments, we have removed the investigation of amiloride sensitivity/potential for these channels (was figures 9 and 10) and removed *del-10* from Figure 8 (was figure 7).

We have edited the text (line 306) to make clear that this is our preliminary screen, from which we identified candidates to investigate in more detail.

We have edited the text (line 330) to take account of the likelihood that these subunits either are not expressed, depend on another subunit to function and/or are not properly localised, noting the examples of ASIC2b and FLR-1/ACD-3/DEL-5, and reiterated this in the discussion (line 563).

We have also added Table 3, showing supporting information for Figure 3 (descriptive statistics and statistical comparison with water-injected controls).

2. Mec-4 and mec-10 data are somewhat concerning. Goodman et al. 2002 and Bianchi et al. 2004 showed that wild type mec-4 produces very small currents when expressed in *Xenopus* oocytes (<0.05 uA). The authors are showing a few data points that are above 0.5 uA,

suggesting that in this case they are reporting oocytes endogenous currents. Wild type *mec-10* does not produce currents when expressed in oocytes, as it functions as an accessory subunit. Yet the authors are reporting several data points with substantial currents. Again, there is concern here that the authors are reporting endogenous oocytes currents.

We agree that these observations, for a small subset of oocytes expressing *mec-4* or *mec-10*, are surprising. However, MEC-10 is not strictly an accessory (i.e. non-pore-forming) subunit. It is homologous to MEC-4 and although it has not been shown to form homomeric channels there is no strong evidence that this would be impossible. So, it seems totally reasonable to report the data rather than ignore it just because it conflicts with expectations.

As our results in this figure show, we do not observe these currents for the water-injected controls or for many other DEG/ENaC subunits, only *mec-4*, *mec-10* and a single oocyte expressing *deg-1*. This argues against the idea that these are endogenous currents; if they are, they are dependent on the presence of *mec-4/mec-10*. Our experimental setup is different to that used in previous papers, so the reason for these currents does indeed merit further investigation, but is outside the scope of this paper.

We have added a section in the discussion noting these currents and the discrepancy with published results (line 567).

We have also altered the axis lengths of the graphs in figure 3, to make it visually more obvious that there are also many data points with very small currents. We have also added Table 3, showing descriptive statistics (median and interquartile range) and statistical comparison with the water control.

3. Why is there one single oocyte that expresses *deg-1* that has substantial current? Is this again endogenous current?

Again, the fact that we do not observe these currents for the many other subunits tested, in large numbers of oocytes, from different batches, or for the water-injected controls, would suggest that this is not an endogenous current. If so, it is a rare occurrence (only one of the 23 oocytes tested for *deg-1*).

As outlined above, we have adjusted figure 3 and added a supporting table to try to make clearer that this is an outlier in a large number of oocytes.

4. Del-9 currents are insensitive to amiloride and non-selective. I have no confidence that these currents are actually due to expression of del-9. There is an endogenous current in *Xenopus* oocytes that has exactly the same kinetics as the currents shown in figure 6 D and which may be pH and zinc sensitive. They also need to keep in mind that this endogenous current is variable from batch to batch of oocytes and that it is more prominent in oocytes that fail to express the protein of interest. The authors need to perform additional experiments in support of their conclusion that the currents reported are indeed del-9. Point mutations that change the channel selectivity or that render the channel nonfunctional might be useful here.

All experiments were performed using at least 3 different batches and, as our results show, the DEL-9 currents are consistently observed across batches. Any bad batches of oocytes were, of course, rejected and not included in any of our experiments. In figure 3, we show a large number of experiments, with oocytes expressing many different channel subunits, that do not exhibit the currents that we consistently observe for DEL-9.

However, to address the referee's concerns, we present 2 additional lines of evidence (Figure 6, together with text starting line 423). First, we injected increasing concentrations of *del-9* cRNA, which showed that current increased with cRNA concentration (at pH_{50} , with the water-injected control oocytes). Second, to render the channel non-functional, we created a deletion mutant of the conserved 'GAS' belt, based on previous research, which abolished acid-evoked currents completely.

We have also discussed in more detail why we have used oocytes and the nature of endogenous acid-activated inward current in *Xenopus* oocytes, referring to previous publications (line 309).

In the discussion, we have also added a paragraph discussing non-selective DEG/ENaCs and the role of subunit compositions and structure of ASICs that are involved in forming a functional channel and that are responsible for ion selectivity.

5. I have similar concerns on *unc-105* currents. Especially the currents shown in figure 7 are not convincing and could be endogenous. Have the authors tried to inject higher concentrations of RNA or record currents earlier or later after the injection? Larger currents, clearly above the level of endogenous current for that batch of oocytes would be more convincing. Analysis of the selectivity and/or point mutation that disrupt selectivity or function would also be needed here.

Thank you for this suggestion, we have now provided panels on ion selectivity (Fig.4C-D) and a panel (Fig.6B) showing concentration-dependent currents of UNC-105h, demonstrating that with increasing concentration of cRNA, currents increase. To further show that the currents are due to UNC-105h activity we generated a non-functional channel by deletion of the conserved 'GAS' belt based on previous research which abolished acid-sensitive currents completely (Fig. 6E, F).

6. Figure 4: the dose response curves have quite large SD and lack a point or two around pH 6.5 that would give more confidence that the estimate of the $\text{pH}_{0.5}$ is correct.

We have now added a new dose-response curve for DELM-1 with more data points (and therefore smaller error bars, Fig. 4).

7. In figure 4 panel A, *delm-1* currents seem to be stimulated by pHs 7.5 and 6 as compared to 8. The authors should discuss this.

We agree that this is an interesting point that we should have discussed. We have added a sentence in the discussion (line 638), putting this in context with previous observations that some modulators show these “dual” effects on different channel subunits.

8. Figure 12 panel A: zinc seems to have a dual effect on ACD-1 depending on the concentration. The authors need to discuss this.

We have discussed this in the light of a similar finding described previously by Jiang et al. 2020 (PMID: 32887365) (line 508). In addition, we have also expanded the discussion as some modulators show these “dual” or “bidirectional” effects on different channel subunits (line 638).

9. Since delm-1 and delm-2 are expressed in the same cells, it would be interesting to determine whether delm-2 regulates delm-1 pH sensitivity.

Absolutely, this is a very interesting question, given the results of Han et al., although RNA seq. data indicate that their expression patterns do not entirely overlap. We have now included the co-expression of DELM-1 and DELM-2 in Figure 4 and discussed co-expression and the interpretation of these results (line 367).

Minor:

1. Figure 2, the yellow circles are hard to see.

Apologies. We have enhanced the yellow circles to make them more visible.

2. Line 314: the term "excitatory" is normally used in reference to a synapse or a current. It might be better to use a different term here.

We agree with the referee, we deleted “excitatory” (line 393).

3. Line 357: "Actual current..." this sentence is not clear. What do the authors mean by "actual current"?

Raw current was referred to as the “actual current” that the oocytes display without baseline subtraction to account for potential leak current. We have edited to “raw” (line 213 and figure legends).

4. Line 391: "other DEG/ENaC subunits.." which ones? The authors need to list them.

This section has been deleted.

5. Line 403: "high concentrations of Zn²⁺". How high? The authors need to provide a range. High is a vague term.

We agree with the referee, we have added the range of the dose-response and the EC₅₀ values for the channels in question (line 498).

6. Line 621: *acd-1* is expressed in the Amphid sheath cells, not socket cells.

Thank you for pointing that out, and apologies for the error. We have corrected this (line 625).

Referee #2:

This manuscript by Kaulich et al. aimed to investigate and characterize potential DEG/ENaC ion channels in *C. elegans*. Given the importance of DEG/ENaC channels in many vertebrate physiological processes and the experimental accessibility of *C. elegans* as a model organism, this study is of interest to a broad readership. However, this reviewer has several concerns which should be addressed.

Major concerns:

1. The entire manuscript lacks statistical analyses of the presented data.

Much of our data is in the form of dose-response curves or I-V curves, so is not suited for meaningful statistical analysis. However, we have included descriptive statistics where appropriate (i.e. reporting the Mean and SD, and N numbers) in line with the Journal's guidelines.

We have added statistical analysis of the results presented in figure 3 (see table 3). We have also now added a new Figure (now Figure 6) in which we used statistical analysis to compare amplitudes of currents and in Figure 4 we used statistical analysis to compare co-expression of DELM-1 with and without DELM-2.

2. Some parts of the manuscript read like a review article. For example, Results, Lines 301-307. The manuscript should be rewritten to meet the formal research article guidelines.

We have rewritten sections of the Results to shorten the description of previously characterised channels and referred to the respective publications instead, and have moved some information to the introduction (for example, the section on previously identified expression patterns) (line 118).

3. Figures 1 and 2 should be placed and described in the results section. Please indicate in the legend of Figure 1 that protein sequences were used to construct the tree. Were the database entries verified? Did the protein sequences match the genomic information available in the

databases (there can be discrepancies between coding sequences and amino acid sequences). Please indicate the used acronyms as well (e.g., lower case letters indicating species of one group).

We have moved figure 1 and 2 and the paragraph referring to them to the Result section. We have added in the legend to figure 1 that protein sequences were used to construct the tree, and indicated the acronyms used, as suggested.

We apologise, at the bottom of the Figure 1 legend it should state "Accession numbers can be found in Table 1." Not Supplementary table 1. We have changed that.

Yes, all data used was verified, the respective references can be found in the main text, we added this information to the methods.

4. I struggle understanding the data presented in Figure 3. Positive current values are shown, indicative of outward currents. This is quite surprising given that vertebrate DEG/ENaCs are cation channels that usually display inward currents. There is also a discrepancy with the data presented in Figure 5, where e.g., ACD-2 shows clear inward currents at -60 mV. Please provide original current traces for the investigated ion channel candidates.

Apologies, we agree this was misleading, these are indeed inward currents. We have replotted the data showing the negative values.

5. Many of the investigated DEG/ENaC channels do not show pH-sensitive currents (Figure 3) and transmembrane currents are within the magnitude of water-injected controls. From these data one cannot conclude that these proteins do not form functional (Acid-sensitive) ion channels. Appropriate controls showing that the proteins are indeed present in the oocyte membranes are needed to support the electrophysiology data.

Yes, we absolutely agree, the absence of current does not mean that the channel is not pH-sensitive, nor can any conclusion be drawn about the localisation on the membrane of the oocyte, and we are not trying to make any more of the negative results than to say we tried to express the channels and did not find currents. Unfortunately, the journal does not allow us to put the negative results in a supplemental figure.

We have added to the results description to make these points (line 330); we have also edited the description of results in Fig 3 to make clear that this is an initial screen from which we selected subunits that merited further investigation (line 305).

6. Methods, Line 193: There appears to be an error regarding the injection volume. I doubt 25 microl would fit into one oocyte.

Yes absolutely, this should be nL! Thank you for pointing that out, we changed it to nL (line 198).

7. For the presented electrophysiology data, please indicate 0 current levels in order to allow justification of inward/outward currents.

All currents are inward currents; we have edited figure 3 to address this issue. In the other figures, all currents are baseline subtracted and drift corrected in the roboocyte software.

8. Results, Line 321-331: I do not fully understand the Erev shift experiments. If extracellular ions are substituted and the ion channel is conducting the substitutes, it is not surprising to obtain a shift in Erev. But does it indicate a "preference" or "selectivity" as stated in the manuscript? It would also be helpful to provide the estimated Nernst potentials for the experimental conditions.

We have rephrased the results and included permeability ratios calculated as previously described (Lynagh et al., 2020) where the ratios, PX/PNa , were calculated using the following Goldman–Hodgkin–Katz equation: $PX/PNa = \exp [(V_{rev,Na} - V_{rev,X})F/RT]$. As the Nernst equation only considers one ion and the internal ion concentration has not been very well described in oocytes. We have included a justification in the Methods (line 217).

9. Figure 8: Please explain how the dose-response curve was plotted, given the activation/inhibition that is dependent on the amiloride concentration.

We agree that this was confusing. We have replaced the figure panel, now plotting two dose-response curves to cover the different concentrations.

10. Many representative traces/figure panels (e.g. of the I/V experiments) are very small and very difficult to read.

We absolutely agree. Apologies. We have deleted two of the I/V figures and edited the third, enlarging the panels.

Minor concerns:

1. Introduction Lines 50-66: Some amphibian ENaCs have a much stronger pH-sensitivity than human ENaCs and there is somewhat conflicting data on the pH-values that activate human ENaCs (e.g. PMID: 31248986). For ASICs, they are not only open or closed, they also show desensitization and a steady state desensitization (e.g. PMID: 35201495). This should be included in the introduction as well.

Thank you for your suggestions, we included them in the introduction (line 78).

2. Table 1: Do all accession numbers refer to Uniprot? The methods suggest that NCBI and wormbase are also used.

Yes, they refer to the Uniprot accession numbers, we have changed the Methods to state this.

3. Methods, Lines 197/198: What was the rationale for filling the electrodes with a mixture of KCl and K-acetate?

The rationale of filling the electrodes with K-acetate in addition to KCl was to reduce the build-up of salt crystals. We have added a sentence explaining this in the Methods (line 203).

4. Methods, Line 205: Please clarify what is meant by "within the error of 210 mOsm". Does this mean that 210 mOsm changes were accepted?

We agree, this was misleading! We clarified this in the Methods (line 212). Including this statement was based on the finding that ENaC block by amiloride is influenced by low osmolarity of the extracellular solution.

5. The reference "Petratou et al., submitted" should be removed - this is not a citable source.

Apologies; this has unfortunately not been submitted to a preprint server, so we cannot cite it yet. We have altered the citation to initials + names, "unpublished observations", in line with the journal's guide for authors, and reduced the number of references we make to the data therein, to address the journal's requirement that unpublished material be referred to sparingly. We have also changed the wording of the Results section where we refer to these observations, to separate our description of these unpublished observations from published work.

6. Table 3: "in" or "ex" are mentioned in the legend but these are not used in the table.

"In" and "ex" are indeed used in the table (next to the pH50s of ACD-5). We have changed excitatory (ex) to activation (act) based on a previous referee comment.

7. There are quite a few statements relating to earlier work by the authors which are not supported by references.

Apologies, Kaulich et al. 2021 (in bioRxiv) was lacking the doi number. However, since this is now published in eLife, we have edited the reference accordingly. We have also edited how we refer to this data throughout the results section, to make clearer which observations are previously published and which are described in this manuscript. The issue of Petratou et al. has been addressed as outlined in response to comment 5 above.

Additional changes:

We have changed the outline of Figure 10 and 11 indicating time of stimulation with pH/Zn solution (and corrected a mix-up in the concentration annotation of the traces).

Dear Dr Walker,

Re: JP-RP-2022-283238R1 "Proton-gated; acid-sensing ion channels Physiological insight into the conserved properties of *Caenorhabditis elegans* acid-sensing DEG/ENaCs" by Eva Kaulich, Patrick TN McCubbin, William R Schafer, and Denise S Walker

Thank you for submitting your manuscript to The Journal of Physiology. It has been assessed by a Reviewing Editor and by 2 expert Referees and I am pleased to tell you that it is considered to be acceptable for publication following satisfactory revision.

The reports are copied at the end of this email. Please address all of the points and incorporate all requested revisions, or explain in your Response to Referees why a change has not been made.

NEW POLICY: In order to improve the transparency of its peer review process The Journal of Physiology publishes online as supporting information the peer review history of all articles accepted for publication. Readers will have access to decision letters, including all Editors' comments and referee reports, for each version of the manuscript and any author responses to peer review comments. Referees can decide whether or not they wish to be named on the peer review history document.

Authors are asked to use The Journal's premium BioRender (<https://biorender.com/>) account to create/redraw their Abstract Figures. Information on how to access The Journal's premium BioRender account is here:

<https://physoc.onlinelibrary.wiley.com/journal/14697793/biorender-access> and authors are expected to use this service. This will enable Authors to download high-resolution versions of their figures. The link provided should only be used for the purposes of this submission. Authors will be charged for figures created on this premium BioRender account if they are not related to this manuscript submission.

I hope you will find the comments helpful and have no difficulty returning your revisions within 4 weeks.

Your revised manuscript should be submitted online using the links in Author Tasks: Link Not Available.

Any image files uploaded with the previous version are retained on the system. Please ensure you replace or remove all files that have been revised.

REVISION CHECKLIST:

- Article file, including any tables and figure legends, must be in an editable format (eg Word)
- Abstract figure file (see above)
- Statistical Summary Document
- Upload each figure as a separate high quality file
- Upload a full Response to Referees, including a response to any Senior and Reviewing Editor Comments;
- Upload a copy of the manuscript with the changes highlighted.

- A potential 'Cover Art' file for consideration as the Issue's cover image;
- Appropriate Supporting Information (Video, audio or data set https://jp.msubmit.net/cgi-bin/main.plex?form_type=display_requirements#supp).

To create your 'Response to Referees' copy all the reports, including any comments from the Senior and Reviewing Editors, into a Word, or similar, file and respond to each point in colour or CAPITALS and upload this when you submit your revision.

I look forward to receiving your revised submission.

If you have any queries please reply to this email and staff will be happy to assist.

Yours sincerely,

Dr Peiyong Fong
Senior Editor
The Journal of Physiology
<https://jp.msubmit.net>
<http://jp.physoc.org>
The Physiological Society
Hodgkin Huxley House
30 Farringdon Lane
London, EC1R 3AW
UK
<http://www.physoc.org>
<http://journals.physoc.org>

REQUIRED ITEMS:

- You must start the Methods section with a paragraph headed Ethical Approval. If experiments were conducted on humans confirmation that informed consent was obtained, preferably in writing, that the studies conformed to the standards set by the latest revision of the Declaration of Helsinki, and that the procedures were approved by a properly constituted ethics committee, which should be named, must be included in the article file. If the research study was registered (clause 35 of the Declaration of Helsinki) the registration database should be indicated, otherwise the lack of registration should be noted as an exception (e.g. The study conformed to the standards set by the Declaration of Helsinki, except for registration in a database). For further information see: <https://physoc.onlinelibrary.wiley.com/hub/human-experiments>.
- The Journal of Physiology funds authors of provisionally accepted papers to use the premium BioRender site to create high resolution schematic figures. Follow this link and enter your details and the manuscript number to create and download figures. Upload these as the figure files for your revised submission. If you choose not to take up this offer we require figures to be of similar quality and resolution. If you are opting out of this service to authors, state this in the Comments section on the Detailed Information page of the submission form. The link provided should only be used for the purposes of this submission. Authors will be charged for figures created on this premium BioRender account if they are not related to this manuscript submission.
- Papers must comply with the Statistics Policy: https://jp.msubmit.net/cgi-bin/main.plex?form_type=display_requirements#statistics.

In summary:

- If $n \leq 30$, all data points must be plotted in the figure in a way that reveals their range and distribution. A bar graph with data points overlaid, a box and whisker plot or a violin plot (preferably with data points included) are acceptable formats.
- If $n > 30$, then the entire raw dataset must be made available either as supporting information, or hosted on a not-for-profit repository e.g. FigShare, with access details provided in the manuscript.
- 'n' clearly defined (e.g. x cells from y slices in z animals) in the Methods. Authors should be mindful of pseudoreplication.
- All relevant 'n' values must be clearly stated in the main text, figures and tables, and the Statistical Summary Document (required upon revision).
- The most appropriate summary statistic (e.g. mean or median and standard deviation) must be used. Standard Error of the Mean (SEM) alone is not permitted.
- Exact p values must be stated. Authors must not use 'greater than' or 'less than'. Exact p values must be stated to three significant figures even when 'no statistical significance' is claimed.
- Statistics Summary Document completed appropriately upon revision.

EDITOR COMMENTS

Reviewing Editor:

We thank the authors for modifying the manuscript based on the comments given following the initial reviews. Both referees have however indicated that there are still concerns regarding some of the data presented that require revision. We

therefore strongly encourage the authors to respond to all points still remaining from the referees and to modify the manuscript accordingly. This includes particular attention to Figure 3 which has comments from both referees. Furthermore, with the data from Fig. 3 now presented in both a figure and tabular format, to avoid repetition a decision should be made on one format of displaying the data, and whichever is chosen - the results of the statistical analyses should be shown. In addition to referee 2's comment regarding "amiloride-insensitive, acid evoked currents", it is of note that the end of the results title (p19, L334 of the updated manuscript), "...degree of ion permeability and amiloride-sensitivity", seems misleading as the data reported in Figures 4 and 5 do not reflect experiments where amiloride was used.

Senior Editor:

We thank you for your attentiveness to the comments raised during previous review. Both expert referees and the Reviewing Editor agree that this revised manuscript is much improved, and that the study has potential to be influential. I understand and appreciate the considerable effort entailed in addressing the initial concerns, and concur with their assessment.

That stated, the revision now also harbors points of in clarity that must be addressed.

Specifically, per the comments of Referee 1, the presentation of data within figure 3E is troublesome, as well as an associated lack of statistical testing for these data, pointed out by Referee 2. The Reviewing Editor concurs with these observations, and moreover suggests that presentation of these data in tabular format (Table 3) are redundant. My recommendation is to incorporate the pertinent descriptive statistics into the figure within the Results, with precise values perhaps indicated within the figure legends.

Another important and recurrent point is the need for further clarification regarding data regarding tests of amiloride sensitivity; please specify how the amiloride-insensitive components were calculated.

In addition, see also point raised by Referee 1 concerning endogenous currents, as they are highly variable between batches. This is related also to comment 5 raised by Referee 2. Please clarify how their appearance is treated in the analysis.

REFEREE COMMENTS

Referee #1:

In this revised manuscript, Kaulich and colleagues performed additional experiments to address the concerns that the reviewers raised during the first round of reviews. While overall the manuscript has improved, I still have some concerns about some of the displayed data, especially the ones shown in figure 3. For example, the authors did a good job with the experiments of figure 6 which convincingly show that *del-9* and *unc-105* currents recorded in the oocytes are indeed originated from the expression of these subunits. However, I still feel that figure 3 panel E show some problematic data. I do agree with the authors that maintaining an open mind is essential in science, but so is using rigor.

There are hundreds of recordings published of both *mec-4* and *mec-10* injected oocytes from at least 2 different groups, none of which shows the currents seen in 3 of the 17 *mec-10* injected oocytes and 4 of the 16 *mec-4* injected oocytes. Of note while in 3 of these 4 *mec-4* injected oocytes the current is inhibited by acidic pH, in one it is activated. This suggests that the some of these recordings might now derive from expression of *mec-4* itself. Because the literature has such an abundance of data showing that currents derived from expression of wild type *mec-4* and *mec-10* are just barely above background if any, the authors should use additional methods to prove that these currents should be included in the graphs and were not merely coming from a sick oocyte. For example, the authors should add amiloride and show that the currents are blocked. To clarify, when talking about endogenous currents, this reviewer not only refers to currents derived from frog ion channels expressed in oocytes but also from membrane leak which could be variable not only from batch to batch but also from oocyte to oocyte. The same applies to the lone *deg-1* oocyte showing substantial current, or even the *del-3* one. Also, it is not clear why most oocytes in all these 4 injections show no current and only a few do. This is highly unusual for the oocytes expression system, unless they recorded way too early after injection, in which case they should allow more time for expression. Overall, I would suggest that panel E figure 3 is deleted. Since the authors report the exact same data in table 3 anyway.

Delm-1 currents shown in table 3 are not statistically different than water-injected oocytes probably because of the low number of oocytes tested for this subunit (only 7). Less clear is why *unc-105* data shown in table 3 are not statistically above the water injected oocytes' currents, especially considering 17 oocytes were tested. Could the authors use the data shown in figure 6 for their table averages?

Minor:

1. Line 309, the parenthesis was never closed.
2. Table 3, there is a formatting error. I believe that +/- should be between the numbers. It would be also useful to add the

number of oocytes tested in the table.

Referee #2:

The authors have revised their initial manuscript and addressed many of my concerns. However, I still have a few comments that should be addressed:

1. The authors have now included statistical analyses of data. But why is there no statistical analysis of the data that is actually presented in figure 3? I.e. currents at pH 7.4 vs. pH 4? This is important since this will clarify if there is indeed an acid-induced activation or inhibition of channel activity.
2. The authors state in their rebuttal letter that all displayed currents are inward currents. Are the UNC-105h currents shown in figure 6 also inward currents (despite being positive)? Please clarify.
3. Line 406: The author state that "To verify whether the amiloride-insensitive, acid-evoked currents of DEL-9-expressing oocytes...". To this point, no experiment with amiloride is reported in the manuscript.
4. Please check references to figure 6 in the text. For example, there is no acid block shown in figure 6B.
5. I still feel the data would benefit from indication of 0 current levels. I understand that normalization/baseline subtraction was employed, but it is very hard to distinguish between stimulus-evoked currents or currents that are stimulus-independent.
6. Procedures for maintenance/culture of *C. elegans* should be reported in the methods section. More information on the employed reporter animals that were generated in this study are needed as well.

END OF COMMENTS

1st Confidential Review

28-Jul-2022

15th September 2022

Dear editors and reviewers,

Thank you very much for your constructive comments on our revised manuscript **JP-RP-2022-283238R1**. Please find below our individual replies to these comments, in purple (line numbers refer to the marked version). We hope you will agree that our manuscript is much improved. In particular, we have addressed the comments regarding Figure 3 by deleting panel E, adding the additional statistical tests requested and deleting the supporting Table 3.

Please note that, in response to our request for further clarification regarding the conflicting comments on Figure 3, we received the following advice from one of the editors:

"I recommend removing the table. The summary statistics for data plotted within Figure 3 (except panel E) then can go into the legends.

Regarding negative data in panel E, I recommend summarizing these data in Results, with values provided within the Statistical Summary document. The Statistical Summary document includes fields for stating the question being tested, as well as indicating n, mean, precise SD values, statistical test performed, and precise p values obtained from the tests. In the Results narrative, the Authors can direct readers to these data within the Statistical Summary Document.

Optional, regarding The Authors' point "While there are multiple possible explanations for the lack of currents (as we outline in the Results), the observation that expression as homomers does not produce reproducible currents is also important information for subsequent investigations, since it implies that other factors/subunits/conditions are likely required for correct localisation and/or functioning of the channel": Indeed, the Authors are positioned to entertain these thoughts more deeply, perhaps in the Discussion."

Best wishes,

Denise Walker.

EDITOR COMMENTS

Reviewing Editor:

We thank the authors for modifying the manuscript based on the comments given following

the initial reviews. Both referees have however indicated that there are still concerns regarding some of the data presented that require revision. We therefore strongly encourage the authors to respond to all points still remaining from the referees and to modify the manuscript accordingly. This includes particular attention to Figure 3 which has comments from both referees. Furthermore, with the data from Fig. 3 now presented in both a figure and tabular format, to avoid repetition a decision should be made on one format of displaying the data, and whichever is chosen - the results of the statistical analyses should be shown.

In response to Referee 1's comments, regarding Figure 3, and further clarification from one of the editors, we have removed panel E from the figure and also removed Table 3, instead referring to the Statistical Summary Table. We have added the additional statistical analysis (comparing pH 4 to pH 7.4) to the Statistical Summary Table.

In addition to referee 2's comment regarding "amiloride-insensitive, acid evoked currents", it is of note that the end of the results title (p19, L334 of the updated manuscript), ".degree of ion permeability and amiloride-sensitivity", seems misleading as the data reported in Figures 4 and 5 do not reflect experiments where amiloride was used.

Apologies for the error. We have edited this to omit the reference to amiloride-sensitivity, because this is explored later in the manuscript.

Senior Editor:

We thank you for your attentiveness to the comments raised during previous review. Both expert referees and the Reviewing Editor agree that this revised manuscript is much improved, and that the study has potential to be influential. I understand and appreciate the considerable effort entailed in addressing the initial concerns, and concur with their assessment.

That stated, the revision now also harbors points of in clarity that must be addressed.

Specifically, per the comments of Referee 1, the presentation of data within figure 3E is troublesome, as well as an associated lack of statistical testing for these data, pointed out by Referee 2. The Reviewing Editor concurs with these observations, and moreover suggests that presentation of these data in tabular format (Table 3) are redundant. My recommendation is to incorporate the pertinent descriptive statistics into the figure within the Results, with precise values perhaps indicated within the figure legends.

We have removed Figure 3E and, following further consultation (see above), we have removed Table 3. We have added statistical testing of the difference between pH 4 and pH 7.4 and added statistics descriptives to what remains of Figure 3 and *p* values to the legend.

Another important and recurrent point is the need for further clarification regarding data regarding tests of amiloride sensitivity; please specify how the amiloride-insensitive components were calculated.

We apologise, this must have been lost during edits, we added how we did the amiloride dose responses and I-Vs to the Methods (line 249).

In addition, see also point raised by Referee 1 concerning endogenous currents, as they are

highly variable between batches. This is related also to comment 5 raised by Referee 2. Please clarify how their appearance is treated in the analysis.

Please see the individual replies to these comments. We have deleted Figure 3E, in response to Referee 1's comments, and made clear in the text that the preliminary screen in Figure 3 shows raw currents and to explain why we did not routinely use amiloride block. We also edited subsequent figure legends to clarify the baseline subtraction, using the pre-stimulus current.

REFEREE COMMENTS

Referee #1:

In this revised manuscript, Kaulich and colleagues performed additional experiments to address the concerns that the reviewers raised during the first round of reviews. While overall the manuscript has improved, I still have some concerns about some of the displayed data, especially the ones shown in figure 3. For example, the authors did a good job with the experiments of figure 6 which convincingly show that del-9 and unc-105 currents recorded in the oocytes are indeed originated from the expression of these subunits. However, I still feel that figure 3 panel E show some problematic data. I do agree with the authors that maintaining an open mind is essential in science, but so is using rigor.

There are hundreds of recordings published of both mec-4 and mec-10 injected oocytes from at least 2 different groups, none of which shows the currents seen in 3 of the 17 mec-10 injected oocytes and 4 of the 16 mec-4 injected oocytes. Of note while in 3 of these 4 mec-4 injected oocytes the current is inhibited by acidic pH, in one it is activated. This suggests that the some of these recordings might now derive from expression of mec-4 itself. Because the literature has such an abundance of data showing that currents derived from expression of wild type mec-4 and mec-10 are just barely above background if any, the authors should use additional methods to prove that these currents should be included in the graphs and were not merely coming from a sick oocyte. For example, the authors should add amiloride and show that the currents are blocked. To clarify, when talking about endogenous currents, this reviewer not only refers to currents derived from frog ion channels expressed in oocytes but also from membrane leak which could be variable not only from batch to batch but also from oocyte to oocyte. The same applies to the lone deg-1 oocyte showing substantial current, or even the del-3 one. Also, it is not clear why most oocytes in all these 4 injections show no current and only a few do. This is highly unusual for the oocytes expression system, unless they recorded way too early after injection, in which case they should allow more time for expression. Overall, I would suggest that panel E figure 3 is deleted. Since the authors report the exact same data in table 3 anyway.

We have deleted Figure 3E, as requested, along with the comments regarding MEC-4 and MEC-10 graphs.

On advice from the editors (see above), who disliked the inclusion of both figure and table, we have also deleted Table 3, since the data shown there is replicated in the Statistical Summary Table.

For the initial screen, we did not verify currents by systematically blocking with amiloride, since family members have been shown to be insensitive or potentiated. We have added a note to the description of the screen to make this clear and to explain why (line 323). We have also edited the text to further emphasise that this is an initial screen for putative acid-sensitive channels, which merited further characterisation (line 333, line 343), and edited the legend to Figure 3 to emphasise that these are raw currents.

Delm-1 currents shown in table 3 are not statistically different than water-injected oocytes probably because of the low number of oocytes tested for this subunit (only 7). Less clear is why unc-105 data shown in table 3 are not statistically above the water injected oocytes' currents, especially considering 17 oocytes were tested. Could the authors use the data shown in figure 6 for their table averages?

Yes, indeed, as we explain, Figure 3 is an initial screen to identify candidate subunits for further characterisation. Many of the UNC-105 currents were very small (as Figure 3 shows), making it difficult to demonstrate significant difference from the water-injected controls. They nevertheless demonstrated a pH dependence and thus merited further investigation, and in figures 4, 6 and 9 you can see our evidence that these are indeed pH-dependent currents, blocked by acid pH and sensitive to amiloride.

A general issue with the acid-inactivated channels is that the channels are open at pH 7.4, i.e. during incubation prior to recording. This appeared to be a particular problem for DELM-1, which exhibits very large currents and appears ultimately detrimental to oocyte health (which incubation with amiloride did not resolve), as the membrane potential will get too high (making it impossible for the roboocyte to clamp, so these unhealthy oocytes were never included in the experiments); the variability likely reflects the shorter incubation time needed to circumvent his problem.

We have now added statistical analysis of the difference between pH4 and pH7.4 currents (to the Statistical Summary, since the editors advised removing Table 3), showing that, for UNC-105, this is significant.

We have also expanded the text describing these results, to point out the small size of UNC-105 currents, and the significant difference between pH 4 and 7.4 (line 337).

We have not added the data from Figure 6 to that of Figure 3, since these are derived from a separate experiment, on different days and using different oocyte batches and controls from those used for the screen in Figure 3, and because this partial dataset would then be duplicated.

Minor:

1. Line 309, the parenthesis was never closed.

Thank you, we have closed the parenthesis (Line 322).

2. Table 3, there is a formatting error. I believe that \pm should be between the numbers. It would be also useful to add the number of oocytes tested in the table.

This is not a formatting issue, it is IQR represented as Q1-Q3, i.e. (-XXX) to (-XXX). Table 3 has been deleted, in response to the editors' request; in the Statistical Summary Table, the column heading for IQR makes clear that this is Q1-Q3. The number of oocytes can also be found in the Statistical Summary Table, and (where relevant) in the legend to Figure 3.

Referee #2:

The authors have revised their initial manuscript and addressed many of my concerns. However, I still have a few comments that should be addressed:

1. The authors have now included statistical analyses of data. But why is there no statistical analysis of the data that is actually presented in figure3? I.e. currents at pH 7.4 vs. pH 4? This is important since this will clarify if there is indeed an acid-induced activation or inhibition of channel activity.

We have now added statistical comparison of pH 7.4 vs. pH4. The descriptors are displayed on Figure 3, with p values in the legend; they can also be found in the Statistical Summary Table (since Table 3 has been deleted in response to the editors' comments, see above).

2. The authors state in their rebuttal letter that all displayed currents are inward currents. Are the UNC-105h currents shown in figure 6 also inward currents (despite being positive)? Please clarify.

Thank you for pointing this out. We acknowledge that the presentation is confusing, as they are baseline subtracted to the pre-stimulus current (i.e. pH 7.4).

We have added a new version of Figure 6, in which we have re-analysed the data the following way: We first baseline subtracted to pre-stimulus current (i.e. pH 7.4, as we have done for all our data) and drift corrected; we then baseline adjust by subtracting the current at pH 5 (where the channel is blocked). We have also added this information to the legend of Fig. 6.

3. Line 406: The author state that "To verify whether the amiloride-insensitive, acid-evoked currents of DEL-9-expressing oocytes...". To this point, no experiment with amiloride is reported in the manuscript.

Apologies for this error. Yes, the amiloride-insensitivity of DEL-9 is now reported later in the manuscript. We omitted "amiloride-insensitive" from this paragraph in line 427.

4. Please check references to figure 6 in the text. For example, there is no acid block shown in figure 6B.

We have deleted this statement (line 436) and made other minor changes to this paragraph. The changes to figure 6 should also, we hope, address the referee's comments.

5. I still feel the data would benefit from indication of 0 current levels. I understand that normalization/baseline subtraction was employed, but it is very hard to distinguish between stimulus-evoked currents or currents that are stimulus-independent.

For most experiments (except where indicated that we present raw current) we used the pre-stimulus current (i.e. at pH 7.4) to baseline subtract, to distinguish between stimulus-evoked and stimulus-independent currents. Therefore, for the acid-activated channels, it is possible to compare the amiloride- (or zinc-) blocked current with the pH 7.4 current; and in figure 8 we show raw currents at pH 7.4, allowing us to compare with amiloride block.

However, we recognise that for the acid-inhibited channels this is less straightforward because at pH 7.4 they are open. This effectively means that “zero” is the highest deviation from baseline.

In an ideal world, we would use a channel blocker, to allow us to compare the pH dependent currents that we observe against a “zero” where the channel is (presumably) completely blocked. But, of course, the amiloride insensitivity/potentialiation/high IC₅₀ of family members complicates this.

We have edited the main text (e.g. regarding Figure 7, line 469) and figure legends, to make the clearer that we baseline subtracted using the pre-stimulus (i.e. pH 7.4) current.

6. Procedures for maintenance/culture of *C. elegans* should be reported in the methods section. More information on the employed reporter animals that were generated in this study are needed as well.

We have now included *C. elegans* growth and maintenance information including references to the standard method of generating transgenic animals in the Methods section (line 171). We have also rephrased the section describing the promoter-fluorophore fusion constructs to provide more detail of their construction and added more detail to the Results (line 515).

END OF COMMENTS

Dear Dr Walker,

Re: JP-RP-2022-283238R2 "Proton-gated; acid-sensing ion channels Physiological insight into the conserved properties of *Caenorhabditis elegans* acid-sensing DEG/ENaCs" by Eva Kaulich, Patrick TN McCubbin, William R Schafer, and Denise S Walker

I am pleased to tell you that your paper has been accepted for publication in The Journal of Physiology.

NEW POLICY: In order to improve the transparency of its peer review process The Journal of Physiology publishes online as supporting information the peer review history of all articles accepted for publication. Readers will have access to decision letters, including all Editors' comments and referee reports, for each version of the manuscript and any author responses to peer review comments. Referees can decide whether or not they wish to be named on the peer review history document.

The last Word version of the paper submitted will be used by the Production Editors to prepare your proof. When this is ready you will receive an email containing a link to Wiley's Online Proofing System. The proof should be checked and corrected as quickly as possible.

Authors should note that it is too late at this point to offer corrections prior to proofing. The accepted version will be published online, ahead of the copy edited and typeset version being made available. Major corrections at proof stage, such as changes to figures, will be referred to the Reviewing Editor for approval before they can be incorporated. Only minor changes, such as to style and consistency, should be made a proof stage. Changes that need to be made after proof stage will usually require a formal correction notice.

All queries at proof stage should be sent to TJP@wiley.com.

Are you on Twitter? Once your paper is online, why not share your achievement with your followers. Please tag The Journal (@jphysiol) in any tweets and we will share your accepted paper with our 23,000+ followers!

Yours sincerely,

Dr Peiyong Fong
Senior Editor
The Journal of Physiology
<https://jp.msubmit.net>
<http://jp.physoc.org>
The Physiological Society
Hodgkin Huxley House
30 Farringdon Lane
London, EC1R 3AW
UK
<http://www.physoc.org>
<http://journals.physoc.org>

P.S. - You can help your research get the attention it deserves! Check out Wiley's free Promotion Guide for best-practice recommendations for promoting your work at www.wileyauthors.com/eeo/guide. And learn more about Wiley Editing Services which offers professional video, design, and writing services to create shareable video abstracts, infographics, conference posters, lay summaries, and research news stories for your research at www.wileyauthors.com/eeo/promotion.

*** IMPORTANT NOTICE ABOUT OPEN ACCESS ***

To assist authors whose funding agencies mandate public access to published research findings sooner than 12 months after publication The Journal of Physiology allows authors to pay an open access (OA) fee to have their papers made freely available immediately on publication.

You will receive an email from Wiley with details on how to register or log-in to Wiley Authors Services where you will be able to place an OnlineOpen order.

You can check if your funder or institution has a Wiley Open Access Account here <https://authorservices.wiley.com/author-resources/Journal-Authors/licensing-and-open-access/open-access/author-compliance-tool.html>

Your article will be made Open Access upon publication, or as soon as payment is received.

If you wish to put your paper on an OA website such as PMC or UKPMC or your institutional repository within 12 months of publication you must pay the open access fee, which covers the cost of publication.

OnlineOpen articles are deposited in PubMed Central (PMC) and PMC mirror sites. Authors of OnlineOpen articles are permitted to post the final, published PDF of their article on a website, institutional repository, or other free public server, immediately on publication.

Note to NIH-funded authors: The Journal of Physiology is published on PMC 12 months after publication, NIH-funded authors DO NOT NEED to pay to publish and DO NOT NEED to post their accepted papers on PMC.

EDITOR COMMENTS

Reviewing Editor:

Dear authors,

Thank you for submitting your revised manuscript. All comments raised by editors and referees have been answered sufficiently.

Senior Editor:

Both Expert Referees, the Reviewing Editor, and I are pleased with the revision of your manuscript. This study not only contributes to deeper understanding of DEG/ENaC channels, but also sets the stage for impactful future work on these channels, using an important model organism. Congratulations on a job well-done, and many thanks for contributing to The Journal of Physiology.

REFeree COMMENTS

Referee #1:

No further comments. I am satisfied with how the authors revised the manuscript.

Referee #2:

The authors sufficiently addressed all my concerns. I have no further comments.

2nd Confidential Review

16-Sep-2022